# Next generation thiazolyl ketone inhibitors of cytosolic phospholipase A₂ α for targeted cancer therapy

Felicity J. Ashcroft [1], Asimina Bourboula[2,3], Nur Mahammad[1], Efrosini Barbayianni[2], Astrid J. Feuerherm[1], Thanh Thuy Nguyen[1], Daiki Hayashi[4], Maroula G. Kokotou[5], Konstantinos Alevizopoulos[6], Edward A. Dennis [7,8], George Kokotos [2,3] ✉ & Berit Johansen [1] ✉

Eicosanoids are key players in inflammatory diseases and cancer. Targeting their production by inhibiting Group IVA cytosolic phospholipase A₂ (cPLA₂α) offers a promising approach for cancer therapy. In this study, we synthesize a second generation of thiazolyl ketone inhibitors of cPLA₂α starting with compound GK470 (AVX235) and test their in vitro and cellular activities. We identify a more potent and selective lead molecule, GK420 (AVX420), which we test in parallel with AVX235 and a structurally unrelated compound, AVX002 for inhibition of cell viability across a panel of cancer cell lines. From this, we show that activity of polycomb group repressive complex 2 is a key molecular determinant of sensitivity to cPLA₂α inhibition, while resistance depends on antioxidant response pathways. Consistent with these results, we show that elevated intracellular reactive oxygen species and activating transcription factor 4 target gene expression precede cell death in AVX420-sensitive T-cell acute lymphoblastic leukemia cells. Our findings imply cPLA₂α may support cancer by mitigating oxidative stress and inhibiting tumor suppressor expression and suggest that AVX420 has potential for treating acute leukemias and other cancers that are susceptible to oxidative cell death.

The polyunsaturated fatty acid arachidonic acid (AA) and its metabolites, eicosanoids, i.e., prostaglandins, thromboxanes, leukotrienes, etc., have attracted considerable interest in relation to inflammatory processes and diseases, including cancer, cardiovascular and autoimmune diseases, and metabolic disorders[1–3]. AA is generated by the action of phospholipase A₂ (PLA₂), which catalyzes its hydrolytic cleavage from membrane glycerophospholipids[4]. PLA₂ enzymes constitute a superfamily of enzymes, divided into many groups and subgroups. Among them, cytosolic Group IVA PLA₂ (cPLA₂α) plays a critical role in the release of AA and is the most well-studied PLA₂ in inflammation[4,5].

The involvement of PLA₂s in various inflammatory conditions renders them attractive targets for the development of medicinal agents. Accordingly, a variety of synthetic PLA₂ inhibitors, targeting the most relevant human PLA₂s, have been reported in the scientific and patent literature[4,6]. Considerable attention has been focused on

[1]Department of Biology, Norwegian University of Science and Technology, Trondheim, Norway. [2]Department of Chemistry, National and Kapodistrian University of Athens, Panepistimiopolis, Athens, Greece. [3]Center of Excellence for Drug Design and Discovery, National and Kapodistrian University of Athens, Panepistimiopolis, Athens, Greece. [4]Department of Applied Chemistry in Bioscience, Graduate School of Agricultural Science, Kobe University, Kobe, Japan. [5]Laboratory of Chemistry, Department of Food Science and Human Nutrition, Agricultural University of Athens, Athens, Greece. [6]Ventac Partners, Ch. du Vallon 4, Yverdon-les-Bains, Switzerland. [7]Department of Chemistry and Biochemistry, University of California at San Diego, La Jolla, CA, USA. [8]Department of Pharmacology, School of Medicine, University of California at San Diego, La Jolla, CA, USA. ✉e-mail: gkokotos@chem.uoa.gr; berit.johansen@ntnu.no

the pharmaceutical inhibition of cPLA$_2$α and some of the most important inhibitors studied in clinical and pre-clinical trials are shown in Fig. 1. Ecopladib and giripladib (1, Fig. 1) are indole-based cPLA$_2$α inhibitors, presenting in vivo efficacy, whose clinical trials for osteoarthritis were terminated, when gastroenterological side effects were observed in a phase II study[7,8]. An inhibitor of related structure, ZPL-5,212,372 (2, Fig. 1), demonstrated excellent efficacy in animal models of airway and skin inflammation and entered trials for topical application[9]. Another potent indole-based cPLA$_2$α inhibitor, ASB14780 (3, Fig. 1), was found to exhibit beneficial effects for the treatment of nonalcoholic fatty liver diseases[10,11]. AK106-001616 (4, Fig. 1) demonstrated strong in vitro inhibition of cPLA$_2$α and considerable in vivo efficacy for inflammation, neuropathic pain, and pulmonary fibrosis[12]. The potent inhibitor pyrrophenone[13] (5, Fig. 1) has been used in various in vitro and in vivo studies, while RSC-3388 (6, Fig. 1) has been shown to sensitize aggressive breast cancer to doxorubicin through suppressing ERK and mTOR kinases[14], and to halt Streptococcus pneumoniainduced polymorphonuclear cells transepithelial migration in vitro, highlighting the importance of cPLA$_2$α in eliciting pulmonary inflammation during pneumococcal infection[15]. Our groups have developed a series of 2-oxoamides as inhibitors of cPLA$_2$α and studied their in vitro and in vivo activities[16]. AX048 (7, Fig. 1) exhibited an antihyperalgesic effect in rats, blocking prostaglandin E$_2$ (PGE$_2$) release evoked by substance P[17]. Among a series of 1-heteroarylpropan-2-ones, compound 8 (Fig. 1) was found to be a highly potent inhibitor of cPLA$_2$α[18]. We have also developed a series of thiazolyl ketones and demonstrated that GK470 (AVX235, 9, Fig. 1) has in vivo anti-inflammatory effects both in a preventative and a curative arthritis model induced by collagen comparable to the reference drugs methotrexate and Enbrel, respectively[19].

In addition to driving inflammatory processes, eicosanoids have established roles in promoting cancer and PGE$_2$ is the best-studied example of this[20]. As such, cPLA$_2$α is now generating interest as a chemotherapeutic target[21], particularly in breast cancer, where *PLA2G4A* overexpression was initially associated with Her-2 positive basal-like subtype and predicted poor response to endocrine therapy[22]. Subsequent studies showed that enhanced choline metabolism in basal-like xenograft models was associated with higher cPLA$_2$α activity[23] and inhibition of cPLA$_2$α with AVX235 inhibited tumor growth as effectively as a dual PI3K/mTOR inhibitor[24]. In line with this, a separate study showed that ASB14780 reduced AA levels and inhibited the growth of triple-negative breast cancer xenografts, dependent on the oncogenic activity of PI3KCA and restricted dietary AA[25]. In other cancers where cPLA$_2$α has prognostic significance e.g.,

lung cancer, acute myeloid leukemia and glioblastoma, synthetic cPLA$_2$α inhibitors have been shown to suppress cancer cell or tumor growth[20,26–30] and in preclinical models of ovarian and cervical cancers were effective sensitization agents for radiation or chemotherapy[31–33]. There is therefore a strong drive to develop more potent, selective, and safe cPLA$_2$α inhibitors for systemic use as targeted anti-cancer drugs and a parallel need to understand the oncogenic events that support cPLA$_2$α dependency in cancer for effective targeting of their use.

In this study, we report the synthesis and characterization of a series of thiazolyl ketone inhibitors targeting cPLA$_2$α, using GK470 (AVX235) as the starting compound. From this series, we identify GK420 (AVX420) as having higher potency and selectivity and investigate its chemotherapeutic mechanism using a cancer cell line screen. Our findings reveal that acute lymphoblastic leukemias exhibit selective vulnerability to AVX420 treatment, and we provide evidence suggesting cPLA$_2$α might promote cancer survival in this context by mitigating oxidative stress.

## Results

### Design and synthesis of the inhibitors

The inhibitors in the present work were designed starting from the structure of the lead thiazolyl ketone GK470 (AVX235), which possesses a ClogP value of 5.96 (calculated using ChemOffice Ultra 11.00). To develop analogs obeying the Lipinski rule[34], the lipophilicity could be reduced by replacing the medium carbon chain of the benzene ring by a chain introducing either a heteroatom such as oxygen or sulfur or a phenyl group (Fig. 2a). Furthermore, introducing a methyl group at the carbon atom between the activated carbonyl and the oxygen atom could potentially improve the metabolic stability.

Phenols 10a–e and thiol 10f were appropriately alkylated to afford esters 11a–j, which further reduced to alcohols 12a–j by DIBALH (Fig. 2b). Oxidation of 12a–j to aldehydes with NaOCl in the presence of TEMPO and subsequent treatment with TBDMSCN and KCN led to cyanohydrins 13a–j. Thiazolines 14a–j were synthesized by reaction of the cyano group of 13a–j with either methyl or ethyl cysteinate, while aromatization to 15a–j was achieved by oxidation with BrCCl$_3$ in the presence of DBU. Finally, the target thiazolyl ketones 17a–j were synthesized after deprotection of 15a–j with HCl/MeOH, followed by oxidation of 16a–j with Dess-Martin periodinane.

An alternative route for the construction of the thiazolyl ring was also explored, in case of lack of commercially available TBDMSCN. Alcohols 12b, g–i were oxidized to the corresponding aldehydes by Dess-Martin periodinane and cyanohydrins 18a–d were synthesized by

**Fig. 1 | Selected inhibitors of cPLA$_2$α.** The structures of selected inhibitors of cPLA$_2$α studied in clinical or pre-clinical trials are presented.

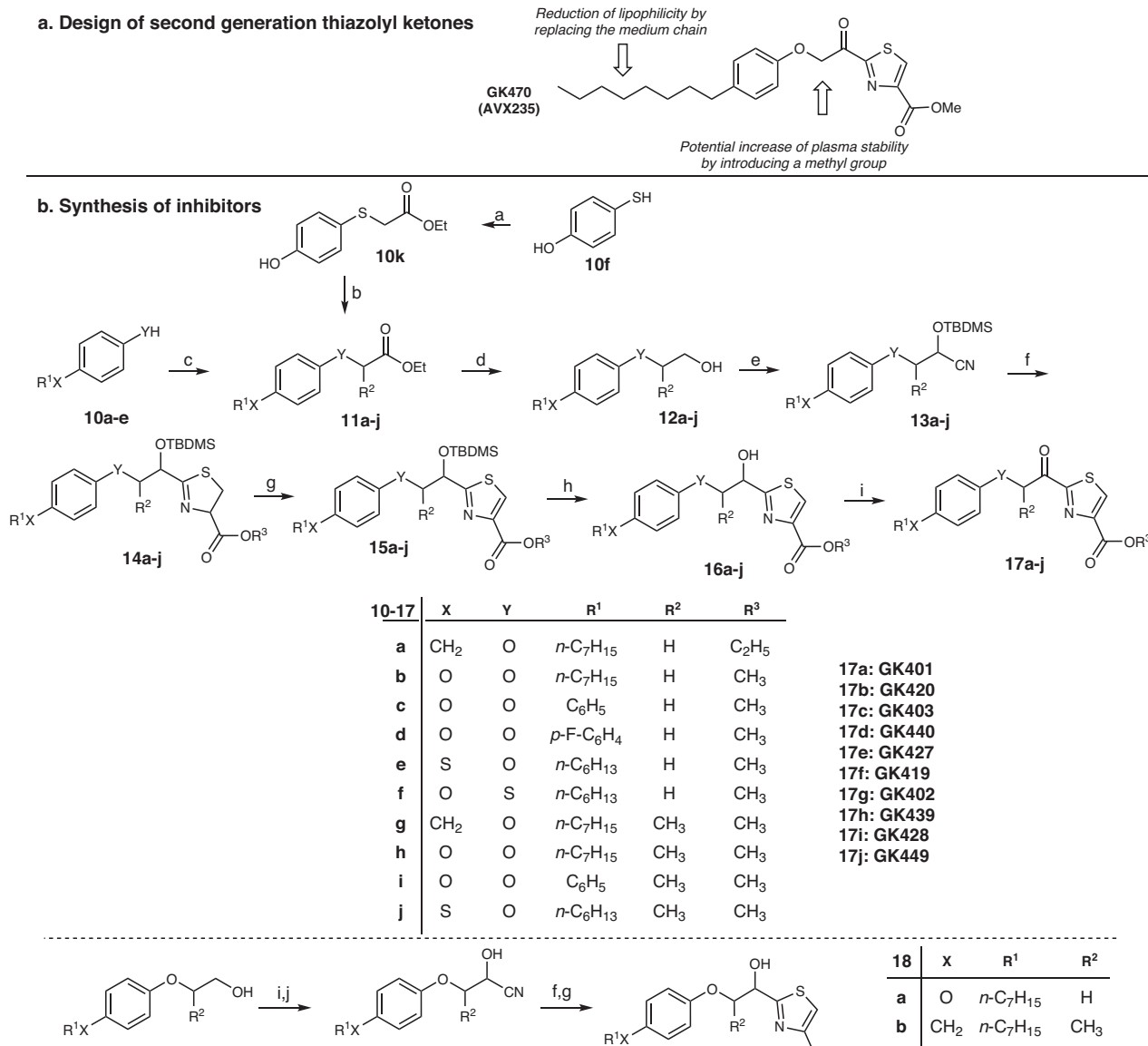

**Fig. 2 | Design and synthesis of thiazolyl ketones. a** Design of second generation thiazolyl ketones. **b** Synthesis of inhibitors. Reagents and conditions: (a) BrCH$_2$COOEt, Et$_3$N, CHCl$_3$; (b) CH$_3$(CH$_2$)$_5$Br, K$_2$CO$_3$, acetone; (c) BrCH$_2$COOEt, or BrCH(CH$_3$)COOEt, K$_2$CO$_3$, acetone; (d) DIBALH, Et$_2$O; (e) i. NaOCl, TEMPO, NaBr, NaHCO$_3$, EtOAc/PhCH$_3$/H$_2$O 3:3:0.5, −5 °C; ii. TBDMSCN, 18-crown-6, KCN, CH$_2$Cl$_2$; (f) HCl.H-L-Cys-OMe or HCl.H-L-Cys-OEt, CH$_3$COO$^-$NH$_4^+$, MeOH; (g) DBU, BrCCl$_3$, CH$_2$Cl$_2$; (h) 4 N HCl/MeOH or 4 N HCl/Et$_2$O; (i) Dess-Martin periodinane, CH$_2$Cl$_2$; (j) i. NaHSO$_3$ 4 N, CH$_2$Cl$_2$, ii. KCN 4 N, H$_2$O/THF 1:1.

treatment with NaHSO$_3$, followed by KCN (Fig. 2b). Cyclization to thiazolines by reaction with methyl cysteinate and subsequent aromatization by treatment with DBU/BrCCl$_3$ afforded compounds **16b, g–i**.

## In vitro inhibition of cPLA$_2$α, GVIA iPLA$_2$ and GV sPLA$_2$

All synthesized thiazolyl ketones were tested for their in vitro inhibitory activity on recombinant human cPLA$_2$α and selectivity over human calcium-independent GVIA iPLA$_2$ and secreted GV sPLA$_2$ using our previously described group-specific radioactivity-based mixed micelle assays[16]. The inhibition results are presented either as per cent inhibition or as $X_I(50)$ values in Table 1. First, the per cent of inhibition for each PLA$_2$ enzyme at 0.091 mol fraction of each inhibitor was determined and then, the $X_I(50)$ values were measured for compounds that displayed greater than 90% inhibition of cPLA$_2$α. The $X_I(50)$ is defined as the mole fraction of the inhibitor in the total substrate

interface required to inhibit the enzyme activity by 50%. For comparison purposes, previously published data for the inhibitor GK470[19] are included.

At first, ethyl ester GK401 showed weaker inhibition in comparison to GK470, indicating that the small methyl ester group is preferable as a substituent on the thiazolyl ring. The introduction of an oxygen atom in replacement of the carbon attached to the phenyl group led to inhibitor GK420, which clearly exhibits more potent inhibitory activity against cPLA$_2$α ($X_I(50)$ 0.0016). However, shortening the carbon chain by one atom (GK419) led to considerable reduction of the inhibitory potency. Thus, an heptyloxy group seems to have the optimum length on the phenyl group, offering potent inhibition and at the same time being acceptable according to the Lipinski rules with ClogP 5.02. Replacing the oxygen atom of GK419 by sulfur (compound GK427) led to increased inhibition of cPLA$_2$α ($X_I(50)$ 0.0010), which is comparable to GK420. Increased inhibition of

**Table 1 | In vitro inhibitory activities and human plasma stability of GK470 analogs**

|  | cPLA$_2$α | | GVIA iPLA$_2$ | GV sPLA$_2$ | Stability in human plasma | |
|---|---|---|---|---|---|---|
|  | % Inhib | X$_I$(50) | % Inhib | % Inhib | 4 h | 20+ h |
| GK470 | >90 | 0.011 ± 0.005 | 86 | 41 | 60.6 | 10.5.5¤ |
| GK401 | 89 |  | 40 | 46 | 69.3 | 16.25¤ |
| GK420 | 98 | 0.0016 ± 0.0002 | 0 | 46 | 63.3 | 18.05 |
| GK419 | 50 |  | 0 | 29 | 82 | 52.7 |
| GK427 | 98 | 0.0010 ± 0.0001 | 7.5 | 44 | 61.3 | 8.05 |
| GK403 | 95 | 0.0014 ± 0.0001 | 34 | 32 | 21.5 | ND¤ |
| GK440 | 100 | 0.0010 ± 0.0001 | 0 | 31 | 21.3 | 0.9* |
| GK402 | 79 |  | 61 | 50 | 79.7 | 47.3¤ |
| GK439 | 97 | 0.0029 ± 0.0004 | 67 | 53 | 59.9 | 12.8* |
| GK428 | 94 | 0.0038 ± 0.0001 | 54 | 36 | 35.9 | 1.65 |
| GK449 | 98 | 0.0017 ± 0.0005 | 67 | 53 | 63 | 18.35* |

The % inhibition of each enzyme is reported at 0.091 mole fraction (% Inhib). X$_I$(50)± stdev is reported for compounds showing >90% inhibition. The stability of the compounds in human plasma is reported as the % of the compound remaining after 4 h (4 h), and either 20 h, 24 h 39 min*, or 25 h ¤ (20+ h) incubation.

**Table 2 | Inhibition of arachidonic acid release**

|  | AA release | |
|---|---|---|
|  | % Inhib | EC$_{50}$ μM (95% CI) |
| GK470 | 67 ± 4 | 0.38 (0.25 to 0.58) |
| GK401 | 53 ± 6 | 0.63 (0.45 to 0.88) |
| GK420 | 97 ± 14 | 0.09 (0.05 to 0.16) |
| GK419 | 71 ± 21 | 0.50 (0.31 to 0.78) |
| GK427 | 56 ± 5 | 0.67 (0.58 to 0.78) |
| GK403 | 57 ± 12 | 0.79 (0.59 to 1.07) |
| GK440 | 88 ± 8 | 0.18 (0.11 to 0.30) |
| GK402 | 23 ± 8 | 6.7 (4.4 to 10.67) |
| GK439 | 80 ± 10 | 0.32 (0.23 to 0.46) |
| GK428 | 12 ± 7 | 9.34 (5.72 to 17.07) |
| GK449 | 58 ± 15 | 0.78 (0.49 to 1.26) |

SW982 synoviocytes were stimulated with IL-1β for 4 h in the absence or presence of 0.2, 1 or 5 μM inhibitor. The mean % inhibition of arachidonic acid (AA) release ± stdev at 1 μM is shown for each inhibitor as well as the calculated EC$_{50}$ values with 95% confidence intervals (CI). Data are from 4 or 3 (GK402) biological replicates.

cPLA$_2$α in comparison to GK470, was also observed for GK403 (X$_I$(50) 0.0014), carrying a phenoxy group instead of the heptyloxy one. Furthermore, the introduction of a fluorine group at the para position of the phenoxy group (GK440) led to potent inhibition of cPLA$_2$α (X$_I$(50) 0.0010). All the analogs mentioned above can be considered selective inhibitors of cPLA$_2$α, because all present minimal inhibition (0-46%) of GVIA iPLA$_2$ and GV sPLA$_2$ at high concentrations (0.091 mole fraction), as shown in Table 1.

Next, we evaluated the thiazolyl ketones carrying an α-methyl group to the activated carbonyl group. In all cases, this small substituent resulted in reduction of the inhibitory potency on cPLA$_2$α. Introduction of the methyl on GK470 (compound GK402) resulted in considerable loss of inhibitory activity. However, α-methyl substituted analogs GK439, GK428 and GK449 retained inhibitory potency (X$_I$(50) 0.0029, 0.0038 and 0.0017, respectively), reduced in comparison to the corresponding analogs without methyl. Again, α-methyl substituted analogs can be considered selective inhibitors of cPLA$_2$α, because all present minimal inhibition (36–67%) of GVIA iPLA$_2$ and GV sPLA$_2$ at high concentrations (0.091-mole fraction) (Table 1).

The stability of the analogs in human plasma was studied over approximately 24 h by LC-MS/MS and the results at 4 h and the final timepoint (20–25 h) are included in Table 1. Inhibitors GK420, GK427 and GK449, which are 7–11 times more potent than GK470, present similar recoveries. On the contrary, GK403 and GK440, although potent inhibitors, were considerably less stable in human plasma. In contrast to our expectations, in this series of thiazolyl ketones, the introduction of a methyl group did not offer higher stability.

### GK420 is a potent inhibitor of cPLA$_2$α in cellular assays

To test the inhibitory activity of the compounds in cells, we measured their effect on the release of AA from synoviocytes stimulated with IL-1β (Table 2). GK420 was the most potent compound, inhibiting 97% of AA release at 1 μM with an IC$_{50}$ of 0.09 μM. Combined with favorable selectivity and stability it was thus chosen for further evaluation. We looked specifically at how it modulated eicosanoid levels in response to stimulation both in whole blood and in isolated peripheral blood mononuclear cells (PBMCs) and checked for off-target effects on AA metabolism in whole blood by also measuring the effect of GK420 on eicosanoid release in the presence of exogenous free AA.

In whole blood, bacterial lipopolysaccharide (LPS) stimulated the release of PGE$_2$, thromboxane B2 (TXB$_2$), and 15-hydroxyeicosatetraenoic acid (HETE). GK420 suppressed their production but had no effect when excess-free AA was included (Fig. 3a) indicating AA metabolism was not affected by GK420. In contrast, the non-selective COX inhibitor indomethacin suppressed the production of PGE$_2$ and TXB$_2$ in the absence and presence of exogenous-free AA. The levels of leukotriene B$_4$ (LTB$_4$), 12-HETE, 5-HETE and 11,12-epoxyeicosatrienoic acid (EET) were not affected by LPS stimulation but increased when exogenous AA was included, irrespective of the presence of GK420 (Fig. S1a).

In PBMCs, the calcium ionophore A23187 stimulated the release of PGE$_2$, TXB$_2$, LTB$_4$, 6-keto prostaglandin F$_1$α (PGF$_1$α), and 12-HETE. LTB$_4$ showed the greatest induction versus unstimulated control in this cell type with an 18-fold increase versus between a 4-to-6-fold increase for the other eicosanoids. GK420 dose-dependently inhibited LTB$_4$, PGE$_2$ and TXB$_2$ with IC$_{50}$ values of 317 nM, 1.1 μM, and 2.2 μM respectively, indicating inhibitory potency might be related to level of stimulation (Fig. 3b). A tendency to suppress both 6-keto PGF$_1$α and 12-HETE was also observed although IC$_{50}$ values could not be calculated (Fig. S1b). In summary, GK420 was found to be a specific inhibitor of cPLA$_2$α, which prevents the stimulated release of AA and production of both COX and LOX-derived metabolites of AA in primary human cells with no discernable inhibition of COX or LOX enzymes.

### GK420 could act as a prodrug generating the corresponding acid, which can bind to the active site of cPLA$_2$α via multiple interactions

To investigate the interactions between GK420 and the active site of cPLA$_2$α, we performed molecular dynamics (MD) simulations followed by docking simulations. However, the significant interactions between GK420 and amino acid residues were not observed in the docking simulations (Fig. S2 and Movie S1). Exploring the human plasma stability of GK420 by a LC-HRMS method, we envisioned that the ester group of this inhibitor might be hydrolyzed by cellular esterases, generating the corresponding acid. Taking advantage of the high mass accuracy of HRMS and following an analytical approach to identify the suspected active inhibitor, we screened for an exact mass corresponding to the acid (M-H$^-$ $m/z$ 376.1224). Gratifyingly, we identified a peak which appears after 1 h of incubation and gradually increases. (Fig. S3). Therefore, we performed docking simulations and MD simulations using the acid form of GK420 (GK420 acid). The interactions between Arg200 and the carboxyl group of GK420 acid were consistent in all docking poses showing lower docking scores (Fig. 4a). The MD simulation further supported strong interaction between

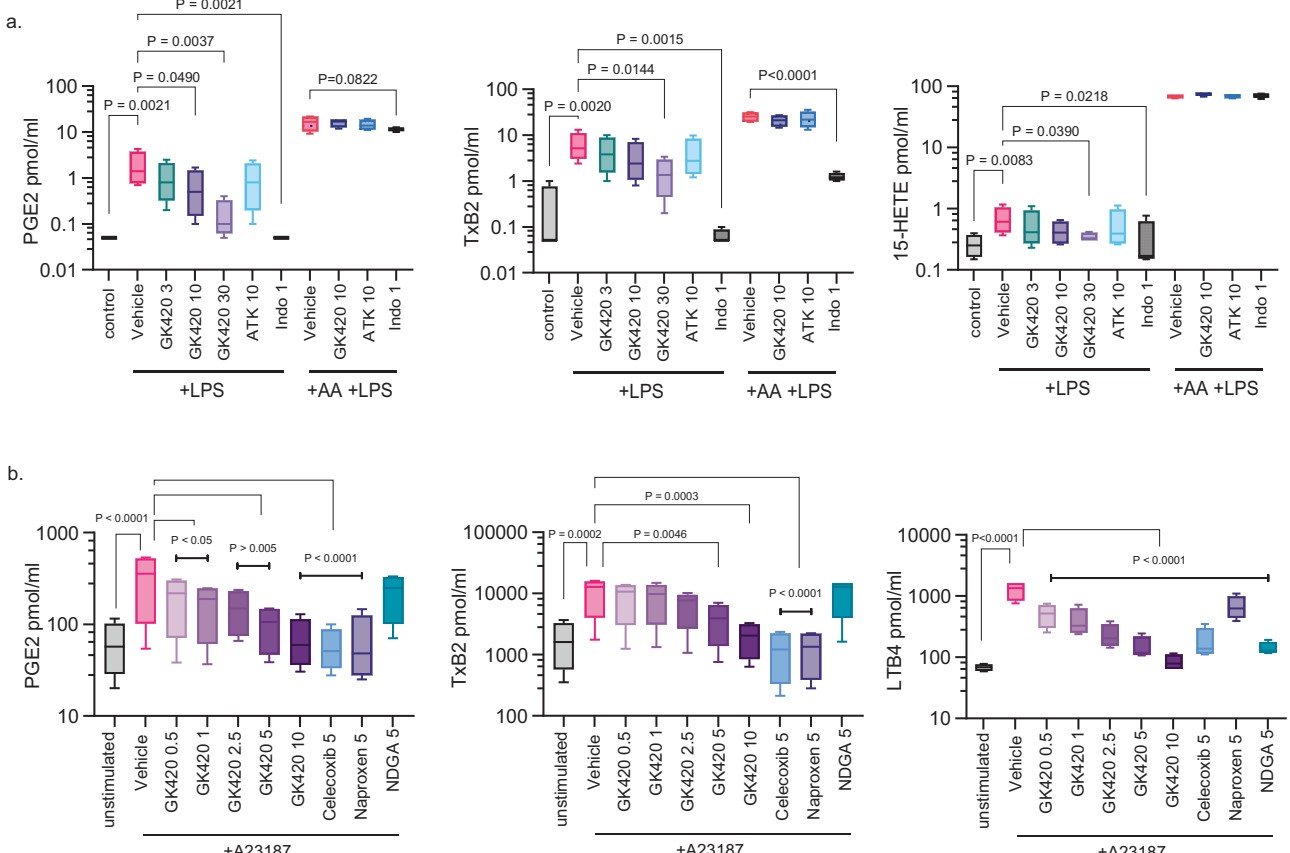

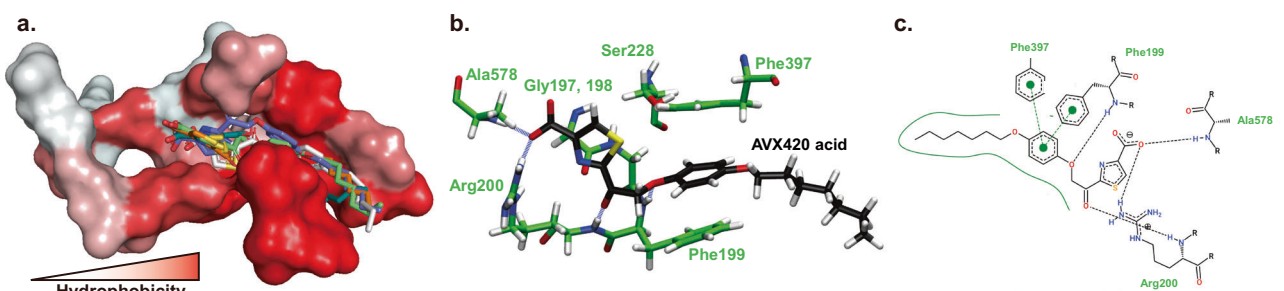

**Fig. 3 | Ex vivo suppression of eicosanoid production by GK420. a** Eicosanoid levels (pmol/mL) in the plasma of whole blood pretreated with vehicle (DMSO), 3, 10 or 30 μM GK420,) 10 μM arachidonyl trifluoromethyl ketone (ATK), or 1 μM indomethacin (Indo) before stimulation with 1 μg/mL LPS for 24 h. In AA + LPS, 5 μM AA was included for the final 30 min. **b** Eicosanoid levels (pg/mL) in the supernatants of isolated human PBMCs pretreated with vehicle (DMSO), GK420, naproxen, celecoxib, or nordihydroguaiaretic acid (NDGA) for 90 min before stimulation with 30 μM A23187 for 15 min. All box and whisker plots show the mean (center line), stdev (box), and min and max values (whiskers) from 4 biological replicates. *P*-values were determined using RM-one way ANOVA with Dunnett's correction for multiple comparisons.

**Fig. 4 | Binding mode of GK420 acid in cPLA₂α binding pocket after MD simulation. a** The docking poses with the five lowest docking scores in the substrate binding site of cPLA₂a are shown. The binding site is shown as a surface and colored based on the Eisenberg scale of the amino acids[84]. **b** The 3D binding mode of the compound after 200 ns representative MD simulation is shown. The blue dashed lines are hydrogen bonds (distance between donor and acceptor 2.6–3.2 Å and donor-hydrogen-acceptor angle >135°) **c** The 2D interaction map of the compound after 200 ns representative MD simulation is shown.

Arg200 and the carboxyl group showing very high occupancy of hydrogen bonds for the entire 200 ns MD simulation (128% in total of side chain NH) (Fig. 4b and Movie S2). In addition, the hydrogen bonds with Ala578 and Phe199 and the pi-pi interactions with Phe199 and Phe397 were also observed and further strengthened the interaction of the compound (Fig. 4c). Of special note, we performed three independent docking and MD simulations using different initial structures, and the interactions with the above residues were reproducible. Based on the above experimental and computational data, we propose that GK420 may act as a prodrug, generating the corresponding acid, which

strongly interacts with the active site of cPLA₂α via multiple critical interactions.

Together, this data was considered sufficient to move GK420 forward to a cell line-based assessment of its anti-cancer potential and GK420 will hereafter be referred to as AVX420.

## cPLA₂α inhibitors target cancers where survival depends on PRC2-dependent transcriptional repression

Targeting specific molecular vulnerabilities in cancer is considered a key approach to reducing the adverse effects of chemotherapy.

However, the events that govern the susceptibility to suppression of the cPLA$_2$α in cancer remain largely unexplored. We therefore took the approach of screening the antiproliferative activity of AVX420 in a panel of 66 cancer cell lines (hereafter referred to as the cancer cell line screen or CCLS) to allow us to investigate molecular and genetic vulnerabilities to the drug. AVX235 and AVX002, a cPLA$_2$α inhibitor of a different chemical class[35], were screened alongside AVX420, such that factors that were found to be in common between compounds could be more reliably considered to be specific to cPLA$_2$α inhibition. All three compounds were broadly effective in the screen, with average IC$_{50}$ values < 20 μM). AVX420 however showed stronger cell line selectivity than either AVX235 or AVX002 (Fig. S4), which manifested in more robust associations with genetic factors in later analyses and may reflect its higher potency.

Comparing the activity of novel compounds against a reference library of known chemotherapeutics has been used historically to predict their mechanism of action[36]. We thus compared the activity profiles of the cPLA$_2$α inhibitors to 121 drugs of known function. In a network tree where linkages with Pearson's correlation of >0.5 are shown, the cPLA$_2$α inhibitors are linked to one another, and AVX002 and AVX235 had no additional linkages. This supports the unique targeted action of these inhibitors. AVX420 had two additional linkages, to cytarabine and melphalan (Fig. 5a). Unsupervised hierarchical clustering of the drug correlations revealed an interesting similarity between the action of cPLA$_2$α inhibitors and multiple histone methyltransferase inhibitors (Figs. 5b, S5), specifically drugs targeting EZH2, the catalytically active component of the polycomb group repressive complex 2 (PRC2). The finding suggests that cPLA$_2$α may play a role in epigenetic silencing of transcription in cancer. A full list of the drugs and their targets, and a table listing the compounds that, based on the hierarchical clustering, showed the most similarity cPLA$_2$α inhibitors can be found in Supplemental Information (Tables S1 and S2 respectively).

Mutations to oncogenes and tumor suppressor genes are major driver of tumor phenotype and can predict the response to classes of chemotherapeutic drugs[37]. We tested whether mutations to 114 validated cancer driver genes represented by at least 3 cell lines in the CCLS were associated with a significant shift in the IC$_{50}$ of the inhibitors. We found that sensitivity to both AVX420 and AVX235 was associated with mutation to the mixed lineage leukemia gene (MLL) and additional sex combs-like 2 (ASXL2), while there were no genes whose mutational status was associated with sensitivity to AVX002. MLL and ASXL2 are both involved in the control of transcription by modulating histone 3 (H3) methylation. MLL, which had the most significant association in each case, codes for a histone lysine [K]-methyl transferases (KMT2A) that activates transcription by tri-methylating H3K4 and antagonizes transcriptional repression by PRC2[38]. Loss of function mutations, such as occur in approximately 4% of all cancers[39], would likely contribute to target gene repression by PRC2. A full list of genes where mutations were associated with sensitivity to AVX420 can be found in Table S3.

Genome-wide gene expression data made available via the cancer cell line encyclopedia[40] was then used to associate drug sensitivity with basal transcript levels. The expression of 18900 genes was available for 56 of the 66 cell lines screened and we correlated this with the activity of each inhibitor. Negative correlations signify association with drug sensitivity and positive correlations with drug resistance. The correlations were first ranked, and relevant signaling pathways associated with inhibitor susceptibility were identified by gene set enrichment analysis (GSEA)[41] against the oncogenic signatures collection in the molecular signature database (MSigDB) (Fig. 6c). In line with the previous results, a strong association between sensitivity to all three cPLA$_2$α inhibitors and high basal activity of PRC2 was indicated by the enrichment of gene sets upregulated by knockdown of components of PRC2, including EZH2. The activity of the mTORC1 pathway, indicated

by enrichment of gene sets upregulated by treatment with the mTORC1 inhibitor rapamycin was also associated with sensitivity to all three inhibitors. The latter result is perhaps unsurprising given the evidence for an interaction between cPLA$_2$α activity and mTOR signaling, particularly in breast cancers[14,25,42]. We additionally found that sonic hedgehog (SHH) and janus associated kinase 2 (JAK2) activation was associated with sensitivity to both AVX420 and AVX235. Of note, the JAK2 inhibitor ruxolitinib clustered along with the cPLA$_2$α and EZH2 inhibitors in the compound activity profiling (Fig. 5b, Table S2) further suggesting involvement of this pathway in drug sensitivity.

Individual cancer-related genes, whose basal expression was associated with sensitivity to cPLA$_2$α inhibition, were identified from a list of 361 genes with known links to cancer. When correlations with an FDR of 5% were considered significant, only associations with AVX420 were identified (Table S4), and in line with the previous results, this list included genes that encode histone methyltransferases including EZH2, as well as DNMT3A, MGA, and other transcriptional repressors (IZKF1, SPEN, and CHD).

While drug sensitivity can correlate to both high and low expression of the target[43], we found no correlation between PLA2G4A expression and sensitivity to the cPLA$_2$α inhibitors either in the cell panel as a whole or after grouping the cell lines based on their tissue of origin (Fig. S6). To date, PLA2G4A transcript levels have only been associated with the response to chemical inhibition in acute myeloid leukemia cells[28], whereas Koundouros et al.[25] showed that it was the stability and activity of cPLA$_2$α protein that predicted response to inhibition. The expression of four genes encoding alternative phospholipase or phospholipase-related proteins did however show a significant negative correlation with AVX420 sensitivity. These were PLCL1, PLCL2, GPLD1, and LPL. None have type II phospholipase activity but are reported to be involved in regulating the use of fatty acids as an energy supply[44–47].

## Protection against oxidative stress defines resistance to cPLA$_2$α inhibition

Individual genes whose basal expression was associated with insensitivity to cPLA$_2$α inhibition were identified from a list of 405 genes with prior association to drug resistance. Again, when positive correlations with an FDR of 5% were considered significant, we only found associations with AVX420, and the genes identified were NFE2L2, CCND1, TSPO and SOCS3. The strongest positive correlation with all cPLA$_2$α inhibitors was NFE2L2, which is the gene encoding Nrf2, a transcription factor that coordinates cellular protection against oxidative stress in both normal and transformed cells[48]. The low number of associations to known drug resistance mechanisms led us to take a second approach that did not use filtering based on prior knowledge. Instead, the activity data from the 121 pre-profiled chemotherapy drugs was used to filter out non-specific associations, and the identified genes are listed in Table S6. Consistent with the first approach there was enrichment of NFE2L2 target genes, but the second approach additionally revealed associations with olefinic, unsaturated fatty acid, and eicosanoid metabolic processes (Fig. 6a). Of note, the gene list included three members of the aldo-keto reductase (AKR) family of enzymes (AKR1C3 AKR1B10, AKR1C2). These targets of the Nrf2 pathway are involved in the detoxification of lipid peroxidation products to protect cells from oxidative stress[49]. These findings indicated that Nrf2 signaling may protect against the effect of cPLA$_2$α inhibition and thus implicated oxidative stress in the chemotherapeutic mechanism of action.

Based on this, we investigated the role of reactive oxygen species (ROS) in cell death induced by cPLA$_2$α inhibition using cell lines that were highly sensitive to AVX420. We reported previously that the cell lines in the CCLS that were derived from hematopoietic cancers were more susceptible to cPLA$_2$α inhibition than those derived from solid tumors[50]. Of these, the cell lines from acute leukemias (CCRF-CEM,

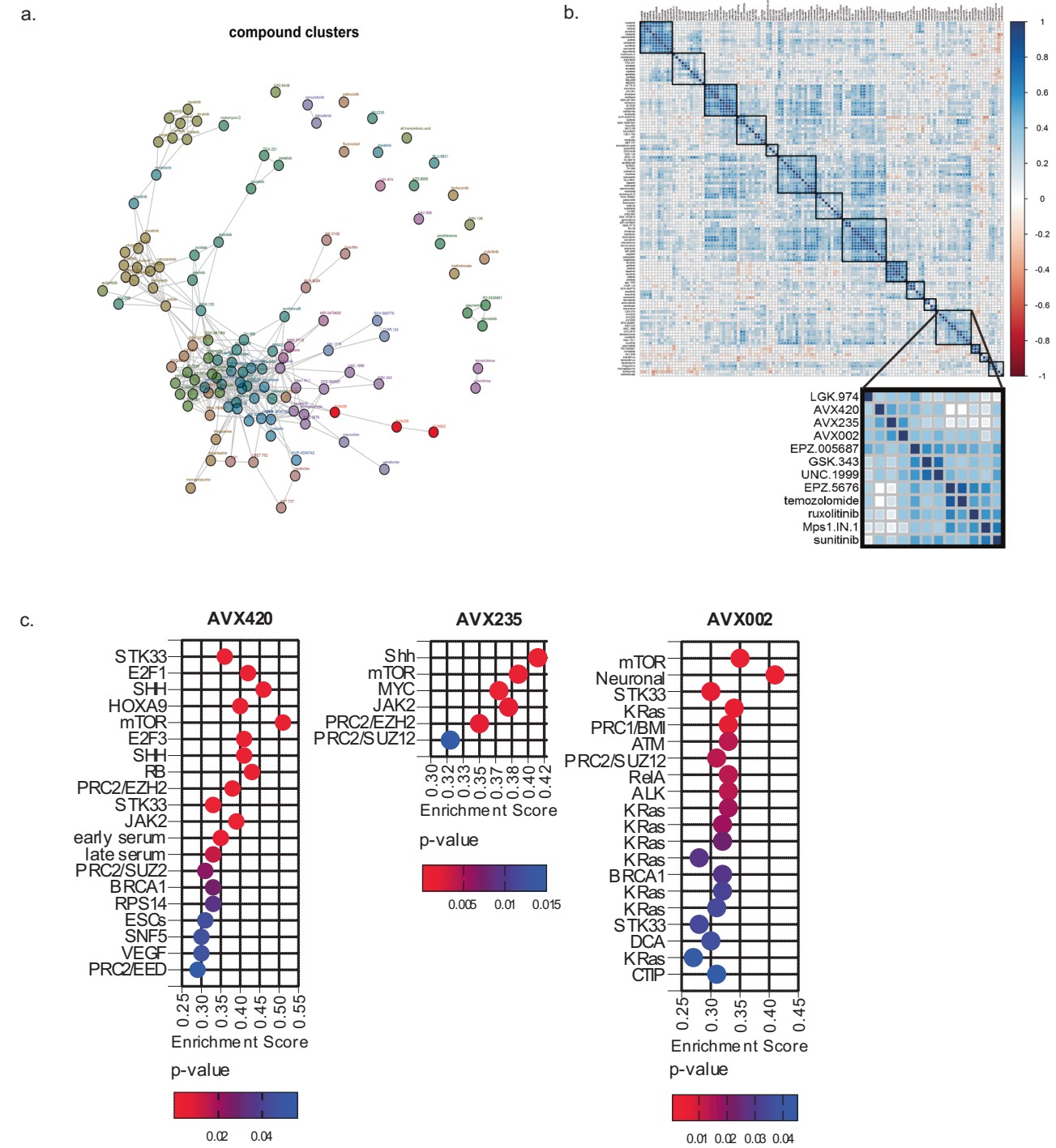

**Fig. 5 | Oncogenic pathways associated with sensitivity to cPLA₂α inhibition.** **a** Network tree showing linkages between related compounds (Pearson correlation coefficient >0.5). Colors indicate to which hierarchical clusters the inhibitors belong. cPLA₂α inhibitors are shown in red. **b** Correlation matrix shows the Pearson correlations corr (**a**, **b**) between all compound pairs. Rows and columns were ordered according to the results of the unsupervised hierarchical clustering to highlight similarities. The color legend on the right indicates matching values for the Pearson correlation coefficients. Reliable clusters, defined by bootstrapping, are outlined in bold. **c** Gene set enrichment analysis based on a ranking of Pearson's correlation with compound activities performed against the Oncogenic signatures collection (C6) in MSigDB. Significantly enriched gene sets (FDR of <5%) are shown.

Jurkat E6.1 and MOLT-4) stood out as being the most sensitive to AVX420 (Fig. S7). We confirmed the strong dose-dependent effect of AVX420 on the viability of Jurkat E6.1 and CCRF-CEM cell lines, which are derived from T-cell acute lymphoblastic leukemias (T-ALL) (Fig. 6b) and further demonstrated that cell lines derived from acute leukemias of a myeloid lineage (MV-4-11 and HL-60) are also susceptible (Fig. S7c). This was in sharp contrast to healthy immune cells (CD3 + T-cells and

PBMCs) where AVX420 had no effect up to a maximum dose tested of 32 μM (Figs. 6c, S8).

Pretreatment with ROS scavengers (glutathione and N-acetyl cystine) rendered the cells completely resistant to cell death induced by AVX420 (Fig. 6d) in contrast to the addition of either AA or a mixture of nine eicosanoids, which did not improve cell viability in the presence of AVX420. Rather, when combined with AVX420,

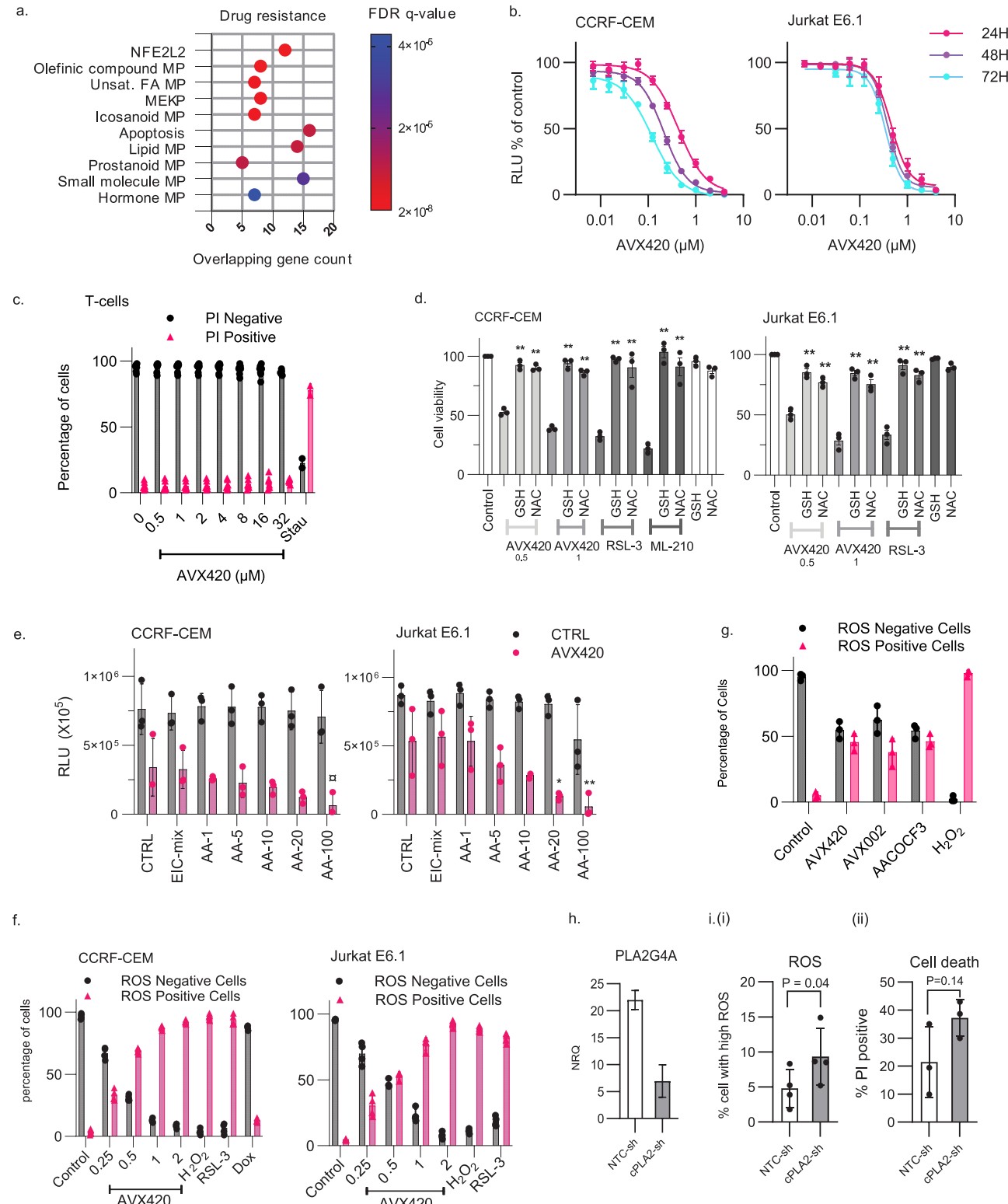

exogenous AA augmented cell death (Fig. 6e). We next measured the effect of cPLA$_2$α inhibition on levels of intracellular reactive oxygen species (iROS) by flow cytometric analysis of ROS-red staining showing AVX420 treatment induced a dose-dependent increase in iROS in Jurkat E6.1 and CCRF-CEM cells (Fig. 6f). The response occurred within 1 h, peaked at 8–12 h, and preceded the loss of cell viability (Fig. S9a, b). Similar to the effects on cell viability, combining AVX420 with exogenous AA did not prevent the increased iROS (Fig. S9c). Two

structurally unrelated inhibitors (AVX002 and AACOCF$_3$) also increased iROS, supporting the specific involvement of cPLA$_2$α (Fig. 6g), and, similar to the effects of pharmacological inhibition, genetic knockdown of cPLA$_2$α using a short hairpin RNA (cPLA2-sh) increased iROS and cell death (Fig. 6h, i). Together, these data implicate cPLA$_2$α in a cancer survival mechanism that is dependent on the suppression of iROS but independent of eicosanoid generation.

**Fig. 6 | Inhibition of cPLA$_2$α causes oxidative stress-dependent cell death in acute leukemia cells. a** The top 10 gene sets from the transcription factor target (C3), gene ontology (C5), and oncogenic signatures (C6) collections in the MSigDB that overlapped with the genes that were associated with resistance to cPLA$_2$α inhibition. **b** Cell viability assessed by CTG assays in CCRF-CEM and Jurkat E6.1 cells. Data are the mean ± stdev of 3 biological replicates. **c** Viability of healthy T-cells measured by propidium iodide (PI) exclusion. Data are the mean ± stdev of 8 biological replicates for 1–16 μM AVX420, 5 for 32 μM AVX420 and 2 for 0.5 μM Staurosporine (Stau). **d** Cell viability measured by CTG assay in cells preincubated with 2.5 mM glutathione (GSH) or 2.5 mM N-acetyl cysteine (NAC) for 30 min before addition of 0.5 or 1 μM AVX420, 0.06 μM RSL-3, or 0.06 μM ML-210 for 24 h. Data are the mean ± stdev of 3 biological replicates. *P*-values were determined by one-way ANOVA with Dunnett's correction for multiple testing and **** indicates *P* < 0.0001. **e** Cell viability measured by CTG assay. CCRF-CEM or Jurkat cells were co-treated with 0.5 μM AVX420 and either arachidonic acid (AA) at concentrations ranging from 1 to 100 μM or a mixture of 9 eicosanoids: 6-keto prostaglandin F$_1$α, Thromboxane B$_2$, prostaglandin F2α, prostaglandin E2, prostaglandin D2, 12(S)- HHTrE, 15(S)-HETE,12(S)-HETE, 5(S)-HETE at a final concentration of 2 ng/mL each (EIC-mix) for 24 h. Data are the mean ± stdev of 3 biological replicates. *P*-values were determined by two-way ANOVA with Sidak's correction for multiple comparisons. **f** Proportion of intracellular ROS-positive cells measured by ROS-red staining and flow cytometry. Cells were treated with indicated concentrations of AVX420 (μM), 1 mM H$_2$O$_2$, 1 μM RSL-3, or 1 μM doxorubicin (dox) for 4 h. Data are the mean ± stdev of 3 biological replicates. **g** Intracellular ROS measurements in CCRF-CEM cells treated with AVX420 (0.5 μM), AVX002 (8 μM), or AACOCF$_3$ (8 μM) for 4 h. Data are the mean ± stdev of ≥3 biological replicates. **h** qPCR analysis of *PLA2G4A* expression. CCRF-CEM cells expressing cPLA$_2$-sh or NTC-sh were separated from non-transduced cells by flow-assisted cell sorting. *PLA2G4A* expression was normalized to *GAPDH* and is shown relative to a non-transduced control. Data are the mean ± stdev of technical duplicates. **i** (i) intracellular ROS or (ii) cell death measured by ROS-red or PI staining in cells expressing cPLA$_2$-sh or NTC-sh using flow cytometry. Data are the mean ± stdev of 4 biological replicates (iROS) and 3 biological replicates (cell death). *P*-values were determined using paired, two-tailed *t*-tests.

## AVX420 initiates an ATF4-dependent but ISR-independent transcriptional program to initiate cell death

Given our findings that the sensitivity to AVX420 was dependent on increased intracellular ROS, we hypothesized that activation of the integrated stress response (ISR) by oxidative damage may be the cause of cell death. The ISR converges on deactivation of the eukaryotic translation initiation factor 2 alpha (eIF2α) to globally suppress translation, with the exception of selected genes (e.g., activating transcription factor (ATF) 4) which activate robust transcriptional changes to restore homeostasis and promote survival or, when oxidative damage is sustained, to initiate programmed cell death[51]. To investigate this, we used a combination of RNA-SEQ and Ribosomal RNA-SEQ (Ribo-SEQ) to measure the global transcriptional and translational response to AVX420 in CCRF-CEM cells. No significant changes in gene expression were seen after 1 hour, 16 genes were upregulated after 6 h (Table 3) and there were 82 differentially expressed genes (DEGs) after 16 h (Fig. 7a, Table S7). Translational effects were determined by calculating the proportion of total transcript associated with the ribosome (ribosomal occupancy (RO)). We did not identify any transcripts with differential RO, indicating that AVX420 did not affect translation. As a positive control we treated cells for 1 hour with the dual mTOR inhibitor Torin-1, which reduced the RO of 20 mRNAs. As expected, GO term enrichment analysis showed these predominantly represented genes coding for ribosomal proteins and translation factors (Tables S8, S9).

Close inspection of the genes upregulated after 6 h treatment with AVX420 revealed that many are established targets of ATF4 (E.g., CHAC1, ADM2, SESN2, SLC7A11, DDIT3, ULBP1, ATF3) and GO term and KEGG pathway enrichment analysis performed on the 82 DEGs at 16 h predicted a response to endoplasmic reticulum (ER) stress (Table S10). While these findings supported the hypothesis that AVX420 initiated cell death via the ATF4 dependent ISR pathway, the absence of translational repression, or selective upregulation of ATF4 translation in response to AVX420, would argue against this, and an alternative mechanism for activation of ATF4 target genes is more likely.

Given our findings that PRC2 activity correlated with sensitivity to cPLA$_2$α inhibition, we instead propose that cPLA$_2$α could be important for epigenetic suppression of ATF4 target genes. In line with this hypothesis, we found a significant overlap between the genes upregulated by AVX420 at 16 h and genes upregulated by the knockdown of BMI1, which is an important component of PRC1 and related to the activity of PRC2 (Fig. 7b). ATF3 and DDIT3 (CHOP) are strong candidates for mediating ATF4-dependent induction of cell death[52–54]. An alternative mechanism could involve L3MBTL2-AS1, an RNA that was upregulated at 6 h, and has been shown to downregulate the expression of L3MBTL2, a transcriptional repressor of DDIT3 (CHOP)[55]. Consistent with the predicted role of mTOR signaling in the response to cPLA$_2$α inhibition, AVX420 upregulated genes computationally overlapped with those upregulated by treatment with the mTOR inhibitor rapamycin. We also found significant overlaps with Nrf2-dependent genes and predicted targets of the transcription factors MafG, Nrf1 and Hes2 (Fig. 7b). Both MafG and Nrf1 are known to cooperate with Nrf2 to regulate the expression of antioxidant response genes[56,57], while Hes2 is a little studied member of the Hes family of transcription repressors downstream of Notch[58]. Given the importance of Notch signaling in T-ALL, interactions between cPLA$_2$α and Hes2 warrant further investigation. We conclude that the transcriptional response to cPLA$_2$α inhibition is compatible with the induction of oxidative stress and loss of mTOR/PRC2-dependent transcriptional repression, which could together activate an ATF4-dependent stress response program leading to cell death.

Given we observed increased iROS 1 h after treatment with AVX420 (Fig. S9a), before both the transcriptional response and cell death (Fig. S9b) and that exogenous AA was ineffective at rescuing these responses (Figs. 5e, S8c), we propose that metabolic alterations could be the cause of the oxidative stress. In addition to its central role in eicosanoid metabolism cPLA$_2$α participates in phospholipid (PL)

## Table 3 | Differentially expressed genes after 6 h of AVX420 treatment

| Gene name | log2FC | Adjusted *p*-value |
|---|---|---|
| CHAC1 | 2.25 | 8.12E-38 |
| L3MBTL2-AS1 | 2.13 | 8.90E-05 |
| SESN2 | 1.59 | 3.07E-28 |
| SLC7A11 | 1.57 | 9.03E-06 |
| ADM2 | 1.49 | 1.65E-10 |
| ENSG00000260498 | 1.44 | 6.37E-03 |
| LINC-PINT | 1.34 | 2.15E-11 |
| VLDLR | 1.27 | 3.68E-10 |
| SLC6A9 | 1.25 | 2.93E-06 |
| GPT2 | 1.24 | 3.60E-05 |
| ULBP1 | 1.2 | 5.06E-04 |
| DDIT3 | 1.15 | 1.13E-18 |
| LY9 | 1.05 | 1.81E-03 |
| ATF3 | 1.05 | 2.20E-08 |
| YAP1 | 1.04 | 9.19E-04 |
| VLDLR-AS1 | 1.02 | 5.37E-05 |

Differential expression analysis using a *P*-value cut-off of <0.01 (after multiple testing adjustments), and a log2 fold change (log2FC) > 1 or <−1.

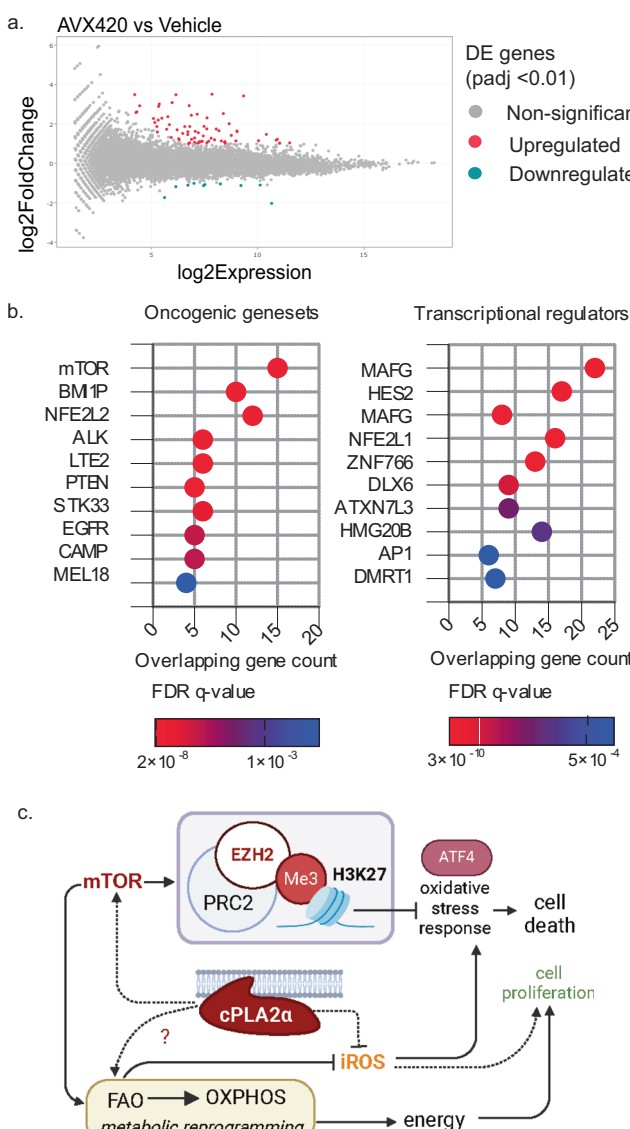

**Fig. 7 | AVX420 activates a transcriptional response to oxidative stress.**
**a** Comparison between fold change and expression in RNA-Seq libraries from CCRF-CEM cells treated for 16 h with 0.5 μM AVX420 or vehicle control. Red and blue dots indicate genes that are expressed differentially with an adjusted *P*-value < 0.01, while grey dots represent genes that are not. **b** Top ten (i) oncogenic, and (ii) transcription factor target (C3) gene sets in MSigDB that overlapped with the genes upregulated after 16 h treatment with 0.5 μM AVX420. **c** Model for the role cPLA₂α in inhibitor-sensitive cancers. Cancers which rely on mitochondrial fatty acid oxidation (FAO) for energy derived from oxidative phosphorylation (OXPHOS) have a high oxidative burden. We propose cPLA₂α is integral to cancer cell survival in this case by inhibiting iROS accumulation and maintaining epigenetic suppression of programmed cell death. Created in BioRender. Ashcroft, F. (2024) https://BioRender.com/r27a486.

remodeling via the Lands Cycle, and is important for the formation of lipid droplets[59,60], which are a significant fuel source in certain cancers including acute leukemias[61,62]. We thus hypothesize that cPLA₂α may support FAO, which in addition to providing energy protects cells against oxidative stress via generation of NADPH[63]. Inhibition of cPLA₂α would then have both epigenetic and metabolic consequences for this cell type (Fig. 7c).

## Discussion

As a therapeutic target, cPLA₂α is well studied in the context of inflammation, and small molecule inhibitors have shown promise for treating diverse chronic inflammatory conditions[64]. Now cPLA₂α is emerging as a cancer target, but mechanistic studies relating cPLA₂α to oncogenesis are limited and methods for predicting the response to cPLA₂α inhibition are not available. Here we describe a series of cPLA₂α inhibitors, which were designed starting from GK470 (AVX235), a molecule with proven efficacy in preclinical models of inflammatory disease and cancer. The lead compound, GK420 (AVX420) was evaluated across multiple experimental platforms and was shown to initiate cancer cell death by the induction of cellular ROS and reactivation of stress response genes. This, along with potential therapeutic uses of AVX420, is discussed below.

Cellular transformation relies on metabolic reprogramming that results from both genetic and epigenetic changes. Fatty acids are recognized as a highly relevant source of energy for cancer cells, particularly under nutrient deprivation, and mitochondrial fatty acid oxidation (FAO) is increased in stem cells and cancer-initiating cells[65,66]. As an energy source, FAO produces high levels of mitochondrial ROS, which can trigger programmed cell death. Adaptive mechanisms for survival are therefore employed to either increase the antioxidant capacity of the cell or prevent oxidative stress-dependent cell death. We found that AVX420 was a potent inducer of ROS-dependent cell death and hypothesize that it targets a hitherto unexplored function of cPLA₂α in the epigenetic adaptation to ROS. Epigenetic changes that act on transcription are a universal feature of cancers and altered histone methylation can lead to aberrant gene silencing or activation by affecting chromatin accessibility[67]. We found correlations between activity of the histone methyltransferase PRC2 and sensitivity to cPLA₂α inhibition, and cPLA₂α inhibitors showed similar anti-cancer activity to inhibitors of the catalytic component of PRC2, EZH2, suggesting overlapping mechanisms of action. Overexpression of EZH2 is a common feature of many solid tumors, where it appears to be important for maintaining cancer stem cell populations and regulating lipid metabolism[68]. It suppresses target gene transcription by trimethylation of H3K27[69] and antagonism is provided by the trithorax group proteins MLL and MLL2, which trimethylate H3K4 to facilitate target gene transcription[38]. Mutations to MLL and MLL2 were also associated with sensitivity to AVX420. We thus propose that cPLA₂α activity regulates transcriptional repression by EZH2 to protect cancer cells that are reliant on fatty acids as an energy source. A direct regulation of epigenetic modifiers by cPLA₂α signaling has to our knowledge not been described to date. However, there is evidence that EZH2 is controlled by mTOR in breast cancer and glioblastomas[70] and can reciprocally promote mTOR activity[71]. mTOR is the central kinase component of two important signaling hubs (mTORC1 and mTORC2) that maintain metabolic homeostasis and regulation of cPLA₂α by mTOR and vice versa have been described before[25,42]. We found that mTOR activation was also associated with sensitivity to cPLA₂α inhibition and AVX420-induced rapamycin-like transcriptional responses. mTOR thus presents as the logical connection between cPLA₂α and epigenetic suppression by PRC2.

Inducing ROS-dependent cell death has become a key approach in cancer biology[72]. An example is the unprecedented success of the combined use of hypomethylating agents e.g., 5-azacytidine and decitabine in combination with the Bcl-2 inhibitor venetoclax for treating acute myeloid leukemia (AML). These agents were shown to induce mitochondrial ROS-dependent cell death and were reported to synergize based on the ability of venetoclax to suppress the Nrf2-dependent antioxidant response[73]. Based on this, venetoclax may also suppress the observed Nrf2-dependent antioxidant response to AVX420, providing synergistic effects, or conversely, by lowering the threshold for cell death, AVX420 could synergize with other ROS-inducing agents. Finally, altered fatty acid metabolism appears as an increasingly important mechanism of chemotherapy resistance[74], including following treatment with venetoclax and hypomethylating agents[75]. AVX420 may be beneficial in instances such as this where

resistance involves increased use of fatty acids as an energy source, especially if epigenetic suppression of cell death is also indicated. Further preclinical experimentation, including in vivo stability and pharmacokinetics, will be critical next steps in the evaluation of the compound for clinical use.

While our study demonstrates the potential of cPLA$_2\alpha$ inhibitors for use in chemotherapy, including the possibility of targeting the vulnerabilities of cancer stem cells and drug resistance in acute leukemias, it also highlights the need to address our limited understanding of the involvement of cPLA$_2\alpha$ in cellular fatty acid metabolism in cancer. This knowledge will be key to fully harnessing the chemotherapeutic potential of cPLA$_2\alpha$ inhibitors, and potent and selective cPLA$_2\alpha$ inhibitors such as AVX420 will also serve us well in this important quest.

## Methods

### General procedures for synthesis of inhibitors

**General method for the synthesis of compounds 11a–j.** To a stirred solution of phenols **10a–e** or **10k** (1.00 mmol) in acetone (10 mL), K$_2$CO$_3$ (3.00 mmol, 415 mg) and ethyl bromoacetate or ethyl 2-bromopropionate or hexyl bromide (1.10 mmol) was added, and the reaction mixture was refluxed for 5 h. Then, the mixture was filtrated over Celite, and the organic solvent was evaporated under reduced pressure. The residue was purified by column chromatography [EtOAc-petroleum ether (bp 40–60 °C), 1:9].

**General method for the synthesis of alcohols 12a–j.** To a stirred solution of esters **11a–j**, (1.00 mmol) in dry Et$_2$O (10 mL), DIBALH (2.5 mL, 2.50 mmol, 1.0 M in hexane) was added at 0 °C under Ar atmosphere, and the reaction mixture was stirred for 2 h at room temperature. Water was then added (5 mL), the reaction mixture was stirred for 30 more minutes and filtrated over Celite. The organic solvent was evaporated under reduced pressure and the residue was purified by column chromatography [EtOAc-petroleum ether (bp 40–60 °C), 3:7].

**General method for the synthesis of nitriles 13a–j.** To a solution of alcohols **12a–j**, (1.00 mmol) in a mixture of toluene (3 mL) and EtOAc (3 mL), a solution of NaBr (1.10 mmol, 110 g) in water (0.5 mL) was added followed by 2,2,6,6-tetramethylpiperidine-1-yloxy free radical (TEMPO) (0.01 mmol, 2.2 mg). To the resulting biphasic system, which was cooled at 0 °C, an aqueous solution of 0.35 M NaOCl (1.10 mmol, 3.1 mL) containing NaHCO$_3$ (3.00 mmol, 250 mg) was added dropwise under vigorous stirring, at 0 °C over a period of 1 h. After the reaction mixture had been stirred for a further 15 min at 0 °C, EtOAc (10 mL) and H$_2$O (10 mL) were added. The aqueous layer was separated and washed with EtOAc (2 × 10 mL). The combined organic layers were washed consecutively with 5% aqueous citric acid (10 mL) containing KI (40 mg), 10% aqueous Na$_2$S$_2$O$_3$ (10 mL), and brine and dried over Na$_2$SO$_4$. The solvents were evaporated under reduced pressure and the residue was used without any further purification.

To a mixture of TBDMSCN (1.00 mmol, 141 mg), potassium cyanide (0.20 mmol, 13 mg) and 18-crown-6 (0.40 mmol, 106 mg), a solution of the aldehyde (1.00 mmol) in CH$_2$Cl$_2$ derived from alcohol, according to the NaBr/TEMPO protocol mentioned above, was added dropwise at room temperature under nitrogen over 30 min. After addition was complete, the reaction mixture was stirred overnight at room temperature. The organic solvent was evaporated under reduced pressure and the residue was purified by column chromatography [EtOAc-petroleum ether (bp 40–60 °C), 0.5:9.5].

**General method for the synthesis of thiazolines 14a–j.** To a stirred solution of the nitriles **13a–j** (1.00 mmol) and CH$_3$COO$^-$NH$_4^+$ (3.60 mmol, 277 mg) in MeOH (4 mL), HCl.H-L-Cys-OMe (3.00 mmol, 515 mg) was added, and the reaction mixture was stirred overnight at

room temperature. The organic solvent was evaporated under reduced pressure and the residue was purified by column chromatography [EtOAc-petroleum ether (bp 40–60 °C), 1:9].

**General method for the synthesis of thiazoles 15a–j.** To a solution of thiazolines **14a–j** (1.00 mmol) in CH$_2$Cl$_2$ (20 mL), BrCCl$_3$ (6.00 mmol, 0.59 mL) and DBU (6.00 mmol, 0.90 mL) were added at 0 °C. The reaction was stirred for 2 h at 0 °C and overnight at room temperature. The organic solvent was evaporated under reduced pressure and the residue was purified by column chromatography [EtOAc-petroleum ether (bp 40–60 °C), 1:9].

**General method for the synthesis of 16a–j.** Compounds **15a–j** (1.00 mmol) were treated with a solution of 2 N HCl in MeOH (10 mL). After TLC indicated complete disappearance of the starting material, the organic solvent was evaporated under reduced pressure and the residue was recrystallized from ether/petroleum ether (bp 40-60 °C).

**General method for the synthesis of thiazolyl ketones 17a–j.** To a solution of compounds **16a–j** (1.00 mmol) in dry CH$_2$Cl$_2$ (10 mL), Dess-Martin periodinane was added (1.50 mmol, 637 mg) and the reaction mixture was stirred for 1 h at room temperature. The organic solvent was evaporated under reduce pressure and Et$_2$O (30 mL) was added. The organic phase was washed with saturated aqueous NaHCO$_3$ (20 mL) containing Na$_2$S$_2$O$_3$ (9.50 mmol, 1.50 g), H$_2$O (20 mL), dried over Na$_2$SO$_4$, and the organic solvent was evaporated under reduced pressure. The residue was purified by column chromatography using petroleum ether (bp 40-60 °C)/EtOAc 7:3 as eluent.

**General method for the synthesis of nitriles 18a–d.** To a solution of alcohols **12b, g–i** (1.00 mmol) in dry CH$_2$Cl$_2$ (10 mL), Dess-Martin periodinane was added (1.50 mmol, 637 mg) and the reaction mixture was stirred for 2 h at room temperature. The organic solvent was evaporated under reduced pressure and Et$_2$O (30 mL) was added. The organic phase was washed with saturated aqueous NaHCO$_3$ (20 mL) containing Na$_2$S$_2$O$_3$ (9.50 mmol, 1.50 g), H$_2$O (20 mL), dried over Na$_2$SO$_4$, and the organic solvent was evaporated under reduced pressure. The residue was purified by column chromatography [petroleum ether (bp 40-60 °C)/EtOAc 8:2], affording the corresponding aldehyde.

To a stirred solution of aldehyde (1.00 mmol) in CH$_2$Cl$_2$ (3.5 mL), an aqueous solution of NaHSO$_3$ 4 N (3.00 mmol, 312 mg) in water (0.75 mL) was added at room temperature. After stirring for 30 min, the organic solvent was concentrated under reduced pressure, water (4.6 mL) was added, and the reaction mixture was cooled to 0 °C. Then, an aqueous solution of KCN 4 N (3.00 mmol, 195 mg) in water (0.75 mL) was added dropwise, and the reaction mixture was left stirring for 16 h. After the completion of the reaction, CH$_2$Cl$_2$ (5 mL) was added and the organic layer was washed with brine (10 mL), dried over Na$_2$SO$_4$, and concentrated under reduced pressure. Purification by column chromatography [petroleum ether (bp 40–60 °C)/EtOAc 6:4], affording the desired product.

Materials, characterization data, HPLC chromatograms of inhibitors **17a–d** and **17g–i**, and $^1$H and $^{13}$C NMR traces are available in the Supplementary Information.

### In vitro PLA$_2$ activity assay

The activities of human recombinant cPLA$_2\alpha$, GVIA iPLA$_2$ and GV sPLA$_2$ were measured using a modified Dole radioactivity-based group-specific mixed micelle assay described previously for measuring the activity of GK470/AVX235[16,19]. Here, cPLA$_2\alpha$ mixed micelle substrate consisted of 400 μM Triton X-100, 95.3 μM 1-palmitoyl-2-arachidonylphosphatidylcholine (PAPC), 1.7 μM arachidonyl-1-$^{14}$C PAPC, and 3 μM PIP$_2$ in a buffer containing 100 mM 4-(2-hydroxyethyl)-1-piperazineethanesulfonic acid (HEPES) of pH 7.5, 90 μM CaCl$_2$, 2 mM dithiothreitol (DTT), and 0.1 mg/mL bovine serum albumin (BSA). GVIA

iPLA$_2$ mixed micelle substrate consisted 400 µM Triton X-100, 98.3 µM PAPC, and 1.7 µM arachidonyl-1-$^{14}$C PAPC in a buffer containing 100 mM HEPES of pH 7.5, 2 mM ATP, and 4 mM DTT. GV sPLA$_2$ mixed micelles substrate consisted of 400 µM Triton X-100, 98.3 µM PAPC, and 1.7 µM arachidonyl-1-$^{14}$C PAPC in a buffer containing 50 mM Tris-HCl of pH 8.0, and 5 mM CaCl$_2$. Inhibitors were initially screened at 0.091 mol fraction (5 µL of 5 mM inhibitor in DMSO) in substrate (495 µL) in technical triplicates. For compounds exhibiting greater than 90% inhibition of cPLA$_2$α, dose-response curves were generated in technical triplicates. The $X_I(50)$ value and stdev values reported are for the entire curve. This analysis was performed in GraphPad Prism v6.

## Plasma stability assays

The metabolic stability of the compounds was determined through in vitro stability assays in human plasma. Solutions of compounds in DMSO (20 mM) were diluted to 5 µg/mL in 50% methanol for infusion into the MS/MS system and spiking of calibration standards and control samples. For the MS detection, positive ionization in ESI mode was used. Good sensitivity, retention and peak shaped were achieved using a gradient with 5 mM ammonium carbamate in water and ammonium carbamate in 90% acetonitrile on a reversed phase column (ACE3AQ). Calibration standards were spiked in a calibration range of 0.5–100 ng/ml. Control samples were spiked at, low, medium, and high concentration levels. The spiked plasma samples were prepared by protein precipitation using acetonitrile containing appropriate internal standard (IS). A lower limit of quantitation (LLOQ) of 0.5 ng/mL was reached for all compounds in human plasma and linearity was good. Samples were prepared for analysis by taking 50 µL of plasma sample and adding 100 µL of IS working solution in acetonitrile followed by vortexing at 2500 rpm (Multi-tube vortexer DVX-2500, Henry Troemner LLC, USA) for 2 min. The samples were centrifuged at 5300 rpm (Heraeus Megafuge 1.0 Heraeus, Germany) for 3 min and the supernatant transferred to the autosampler. Recovery was determined by comparing the prepared matrix samples (protein precipitation) with a direct sample (analytes spikes in 67% acetonitrile = injection solution) such that the recovery of the sample preparation and the matrix effects were determined together. Two or three samples were each measured in technical triplicates to give the final % recovery.

## Cellular arachidonic acid (AA) release assay

The human synovial sarcoma cell line SW982 (ATCC, London, UK) was used to measure AA release. The cells were maintained in Dulbecco's Modified Eagle Medium (DMEM) supplemented with 10% FBS, 0.1 mg/mL gentamicin, and 0.3 mg/mL L-glutamine at 37 °C with 10% CO$_2$, and experiments were performed at 3 days post-confluence. To measure AA release, the cells were labeled overnight with $^3$H-AA (0.4 µCi/mL) in serum-free DMEM and then washed twice with PBS containing fBSA (2 mg/mL) to remove unincorporated radioactivity. The inhibitors, or DMSO vehicle control, were subsequently added to the cells for 1 hour prior to stimulation with 10 ng/mL IL-1β for 4 h. Supernatants were collected and centrifuged at 300 × $g$ for 5 min to clear them of detached cells and adherent cells were dissolved in 1 N NaOH. The $^3$H-AA was assessed by liquid scintillation counting (LS 6500 Multi-Purpose Scintillation Counter, Beckman Coulter, Inc., USA) and the results are given as the inhibition of released $^3$H-AA in the supernatants relative to total $^3$H-AA incorporated into the cells, as described[76]. EC50 values were calculated by fitting the % inhibition data to a three-parameter dose-response curve in GraphPad prism v10.4.0. Experiments were performed in technical triplicates and repeated at least three times.

## Analysis of eicosanoid release from whole blood and peripheral blood mononuclear cells (PBMCs)

This research was performed in accordance with the Declaration of Helsinki. Blood was collected from healthy donors at St. Olavs Hospital HF with their informed consent, as approved by the Regional Ethical Committee of Mid-Norway; #2016/553.

Aliquots of 0.4 mL of heparinized whole blood were mixed with 1.6 mL of serum-free RPMI cultivation medium with- or without indomethacin (# I7378, Sigma-Aldrich), GK420, or arachidonyl trifluoromethyl ketone (ATK) (Cayman #62120), prior to the addition of lipopolysaccharides (LPS) (1 µg/mL) to activate the cPLA$_2$α enzyme (LPS was from E. coli 026:B6 γ-irradiated, #L2654, Sigma-Aldrich). Exogenous arachidonic acid (5–10 µM) (#3555, Sigma-Aldrich) was added after 23.5 h, for the final 30 mins of the 24 h stimulation. Following treatment (24 h, 37 °C, 5% CO$_2$), the diluted blood was centrifuged (5992 × $g$ for 10 min at 4 °C) to isolate the plasma from the cell fraction. Plasma samples were stored at −80 °C until analysis. Samples were analyzed at the Lipid Maps Consortium, UCSD according to the standard methods. 200 µL of sample was subjected to solid phase extraction of the eicosanoids using the StrataX solid phase extraction columns from Phenomenex and analyzed by UHPLC-MS/MS using full standards mix.

PBMCs were isolated from buffy coats using Lymphoprep as described by the manufacturer (STEMCELL Technologies, Vancouver, Canada). Red blood cells were lysed and removed using RBC lysis buffer (Roche, Basel, Switzerland) and the remaining cells were resuspended in 10 mL of RPMI-1640 supplemented with 5% FBS. The TC20 automated cell counter (BIO-RAD, Hercules, California, USA) was used to count the number of cells and check the viability by staining with trypan blue (0.4%) (NanoEnTek, Waltham, MA, USA). For experiments, 5 × 10$^5$ cells were seeded in 0.5 mL in 24 well plates with RPMI medium supplemented with 5% FBS, 0.3 mg/mL glutamine, and 0.1 mg/mL gentamicin. Cells were preincubated with inhibitors for 2 h before incubating with 30 µM Ca$^{2+}$ ionophore A23187 for 15 min to activate cPLA$_2$α. After incubation, the supernatant was obtained by centrifugation at 5992 × $g$ for 10 min at 4° and stored at −80 °C until assayed for PGE$_2$, LTB$_4$, TxB$_2$, 6-Keto PGF$_{1α}$, and 12S-HETE using enzyme-linked immunosorbent assays (ELISAs) (Cayman #514435, Cayman #10009292, Cayman #501020, Cayman #15210, and Enzo Lifesciences #ADI-900-050). After an overnight hybridization, the enzymatic conversion of substrate was read at OD420 nm using a Cytation 5 multimode platereader (Biotek Instruments, Winooski, VT, USA). Standard curves fit using 4-parameter logistic regression in myassays.com were used to calculate eicosanoid concentrations.

## Molecular docking and molecular dynamics (MD) simulations

The structure of AVX420 and AVX420 acid were sketched and optimized using Maestro software (Schrödinger, Inc., NY, USA), and the docking simulation was performed using Glide software (Schrödinger, Inc., NY, USA). For the ligand docking, a scaling factor of 0.8 and a partial charge cutoff of 0.15 were employed[77]. The catalytic domain (AA 144 to 721) of cPLA$_2$α based on the crystal structure (PDB ID: 1CJY) was employed for the simulations[78]. The top-scored protein-ligand complexes from the docking were placed onto a lipid bilayer composed of 1-palmitoyl-2-oleoyl-glycero-3-phosphocholine (POPC) by Membrane Builder, which is implemented in the CHARMM-GUI[79,80]. The enzyme-membrane models were solvated with TIP3P water and neutralized with 150 mM sodium chloride (NaCl). The MD simulations were carried out using NAMD 3.0 software and the CGENFF and CHARMM36m force field and parameters[77,81]. The 200 ns production run was performed using the NPT ensemble after the 200,000-step energy minimization and the 20,000-step equilibration. The temperature and the pressure were maintained at 310 K and 1.01325 kPa, respectively, during the production run. A time step of 2 fs was used with the SHAKE algorithm. The 2D binding mode was analyzed and visualized by PoseView implemented in Proteins Plus (https://proteins.plus/)[82].

## Cancer cell line screening

Screening of compound activities in 66 cancer cell lines was carried out at the Netherlands Translation Research Center. Briefly, proliferation assays were performed 72 h after compound addition using ATPlite 1step reagent (Perkin Elmer). $IC_{50}$s calculated by non-linear regression analysis were used as a measure of compound activity for investigating associations with genetic factors. A complete description of the laboratory and data analysis methods are available in the Supplementary Information file (Supplemental Methods: Cancer Cell Line Screening).

## Cell culture experiments

**Reagents and chemicals.** Cell culture media RPMI 1640 (#R0883), Gentamycin (#G1397), dimethyl sulfoxide (DMSO) (#2650), A23187, naproxen, celecoxib, and lipopolysaccharide (LPS) were purchased from Sigma-Aldrich (St. Louis, MO, USA). Fetal Bovine Serum (FBS) (#10270106) was purchased from Thermo Fisher Scientific (Waltham, Massachusetts, USA). L-glutamine (#17-605E) was purchased from Lonza Pharma & Biotech (Hochbergerstrasse, Basel, Switzerland). Nordihydroguaiaretic acid (NDGA) and Arachidonyl Trifluoromethyl Ketone (ATK) (#62120-10) were obtained from Cayman Chemicals (Ann Arbor, MI, USA). AVX002[35] and AVX235[19] were synthesized by Synthetica AS, Oslo. AVX002, AVX235, and AVX420 were stored at −80 °C in DMSO.

## Cell culture and plating

CCRF-CEM (# CCL-119) and Jurkat E6.1 (#TIB-152) cell lines were purchased from American Type Culture Collection (ATCC, Manassas, VA, USA) while HL-60 and MV-4-11 cells were a kind gift from Prof. Bjørn Tore Gjertsen (University of Bergen, Norway). All cell lines were confirmed to be free of mycoplasma contamination. The cell lines were not authenticated. MV-4-11 were cultured in Iscove's Modified Dulbecco's Medium (Sigma, I3390). CCRF-CEM, Jurkat E6.1, and HL-60 cells were cultured in RPMI 1640 Medium RPMI-1640 medium (Sigma, R5886). Medias were supplemented with 10% FBS (heat-inactivated), 2 mM L-glutamine, 0.1 mg/mL gentamycin, and 10 % FBS (complete media) and kept in a humidified incubator at 37 °C with 5% $CO_2$. Cell lines were tested to ensure the absence of mycoplasma. Cell viability was measured by trypan blue staining, and for experiments, the cells were seeded at a density of 200,000 live cells/mL in 24-well flat-bottom plates. Experiment using CCRF-CEM and Jurkat E6.1 cells were carried out in complete media while experiments using MV-4-11 and HL-60 cells were carried out in media supplemented with 4% heat-inactivated FBS.

## Genetic knockdown of cPLA₂α

Dharmacon SMARTvector lentiviral plasmids for expression of shRNA targeting PLA2G4A (# V3SH11240; gene target sequence: ACAGTGGGCTCACATTTAA), or non-targeting control (# VSC11707; sequence: CACACAACATGTAAACCAGGGA) under control of hCMV promoter and with a TurboGFP reporter were purchased from Horizon Discovery. shRNA expression plasmid, psPAX2, and pMD2 were transfected into human embryonic kidney (HEK)-293 T cells maintained in DMEM supplemented with 10% FBS, 2mM L-glutamine and 500 μg/mL Penicillin-Streptomycin at 37 °C in 5% $CO_2$ using Genejuice (Merck). The media was replaced after 18 h and lentiviral-containing supernatants were harvested after 24 and 48 h, filtered (0.45 μm) and either used fresh or stored at −80 °C.

For analysis of iROS and cell viability, CCRF-CEM were seeded in 24-well plates at $8 \times 10^5$ cells in 100 μL together with 100 μL lentiviral supernatant and 1.25 μg/mL polybrene (Merck) or with polybrene alone (control) and incubated for 6 h. Then, 0.8 mL media was added, and cells were incubated for a further 72 h before analysis. The % of GFP positive cells was 3.0 ± 2.0 for cPLA₂-sh, and 8.7 ± 4.8 for NTC-sh.

This population was gated, and intracellular ROS or PI staining was analyzed using flow cytometry as described below.

To confirm PLA2G4A knockdown, the transduced cells were separated based on GFP expression using flow assisted cell sorting. In this case $1 \times 10^6$ CCRF-CEM cells were seeded into T25 flasks in 1 mL media and incubated together with 1 mL lentiviral supernatant and 1,25 μg/mL for 6 h, before addition of 8 mL media. After 72 h, the transduced cells were isolated by flow-assisted cell sorting (MolGen Sony cell sorter SH800) and total RNA was isolated by Qiagen RNeasy kit according to the manufacturer's instructions. cDNA was synthesized from RNA by using qScript Ultra SuperMix (Quanta bio). Quantitative real time-PCR was carried out by Lightcycler ®96 instrument (Roche) using the PerfeCta SYBR Green FastMix (Quanta bio) with primers against PLA2G4A (Fwd 5′-CATGCCCAGACCTACGATTT; Rev 5′-CCCAATATGGCTACCACAGG) and GAPDH (Fwd 5′-CATCAA-GAAGGTGGTGAAGCAG, Rev 5′-TGTAGCCAAATTCGTTGTCATACC). Fluorescence versus time recorded in Lightcycler v1.1.0.1320 were exported to LinRegPCR (v2021.2) to calculate PCR efficiencies and Ct values and the relative expression was calculated using the Pfaffl method[83].

## Cell viability assays

For plate-based assays, 10,000 cells were seeded in ≥3 technical replicates in 100 μL growth media. After the indicated treatments, cell titer glo (CTG) or resazurin assays were carried out according to the manufacturers' guidelines. For CTG assays, luminescence was determined 10 min after reagent addition. For resazurin assays, fluorescence at 544/590 nm was measured 2 h after reagent addition. Data were acquired using the Cytation 5 cell imaging multimode reader (Biotek Instruments, Winooski, VT, USA).

Propidium iodide exclusion was used to assess cell death by flow cytometry. After treatment, the cells were washed once with PBS and stained with 1.4 μg/mL propidium iodide (PI) for 5 min before quantification of PI negative and positive cells using a CytoFLEX flow cytometer (California, United States). An example of the gating strategy to assess PI staining can be found in Fig. S8c.

## ROS detection

After treatment, cells were stained with cellular ROS Assay Kit (Cat. No. ab186029; Abcam, Cambridge, United Kingdom) following the manufacturer's instructions, and ROS-positive cells were quantified by flow cytometry using either the NovoCyte flow cytometer, and Novoexpress Software v1.5.0 from Agilent Technologies (Santa Clara, USA) or the CytoFLEX flow cytometer and CytExpert analysis software v 2.5 from Beckman Coulter (California, United States). An example showing the gating strategy used to assess ROS-Red staining is found in Fig. S9d.

## RNA sequencing and ribosome-associated RNA-sequencing

CCRF-CEM cells seeded at 200,000 viable cells/mL were treated with AVX420 (0.5 μM) or DMSO (0.1%) and harvested after 1 hour, 6 h, or 16 h. Torin-1 (100 nM) treatment was harvested after 1 h. The flasks were placed on ice and the cell suspensons were transferred to 50 mL conical tubes for centrifugation (405 × g, 4 °C, 5 min). Cell pellets were washed once in 10 mL of chilled Dulbecco's (D) PBS and resuspended in 5 mL of chilled DPBS for UV crosslinking using 254 nm UV light with an energy setting of 400 mJoules/cm² (UVP CL-1000 Ultraviolet Crosslinker). The cells were collected and centrifuged at 405 x g for 5 min at 4 °C. The supernatant was discarded, and the cells were suspended in 1 mL of DPBS before centrifugation at 405 x g for 5 min at 4 °C. Resulting cell pellets were snap-frozen in liquid nitrogen and stored at −80 °C. RNA-Seq and ribosome-associated RNA(Ribo)-Seq was performed by Eclipsebio (eclipsebio.com). Protocols are available in the Supplemental Methods section.

## Data analysis software

Unless stated otherwise, GraphPad Prism v10 was used for graphing and statistical analysis.

## Reporting summary

Further information on research design is available in the Nature Portfolio Reporting Summary linked to this article.

## Data availability

The RNA-seq dataset that was generated in this study is available in the GEO repository with access code GSE251999. Source data for figures are provided with the paper. Source data are provided with this paper.

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

## Acknowledgements

This study was funded by a research sponsorship agreement between Coegin Pharma (former Avexxin) and Norwegian University of Science and Technology, supported by grant NFR235344 from the Norwegian Research Council (NFR). Work on this manuscript by EAD was supported by NIH grant R35 GM139641. We would like to thank Banita Satpathy, Randi Røsbak, and Sandra Kristiansen for technical support and Dr Magne Børseth and Dr Samah Elsaadi for helpful discussions and advice. We would also like to thank Eclipsebio (https://eclipsebio.com/) for their support in the processing and analysis of eRibo Count experiments. Furthermore, we are thankful to Coegin Pharma that supported the collaboration.

## Author contributions

B.J., G.K., and E.A.D. initiated the project; B.J., G.K., E.A.D., A.J.F., K.A., and F.J.A. designed and supervised the project; F.J.A., N.M., E.B., A.B., A.J.F., T.N., D.H., and M.G.K. did experiments and analyzed and prepared data; the manuscript was written by F.J.A., G.K., N.M., A.J.F., K.A., E.A.D., and B.J.

## Funding

 Olavs Hospital - Trondheim University Hospital).

## Competing interests

A.J.F., B.J., F.J.A., E.A.D., and G.K. are shareholders, while F.J.A. and N.M. were employees of Coegin Pharma at the time of results generation. The other authors declare no conflict of interest. The funders had no role in the design of the study, in the collection, analyses, or interpretation of data, in the writing of the manuscript, or in the decision to publish the results.
