## [Transparent Peer Review file · Nature Communications]

Next Generation Thiazolyl Ketone Inhibitors of Cytosolic Phospholipase A2 α for Targeted Cancer Therapy

Corresponding Author: Professor Berit Johansen

Version 0:

Reviewer comments:

Reviewer #1

(Remarks to the Author)

Recommendation: Publish in Nature Communications after major revisions.

Comments:

Ashcroft et al. reported a new class of thiazolyl ketone-based inhibitors targeting cPLA2 α and explored their potential mechanism for chemotherapy. The following issues might be helpful to the authors:

1. The acquisition of crystal structures of protein-ligand complexes is a crucial aspect in the process of structure modification, as it provides essential binding information. Unfortunately, this study is deficient in the availability of pertinent data pertaining to crystal structures.
2. The directing binding of AVX420 with cPLA2 α was not characterized.
3. In Scheme 1A, the mention of a "potential increase in plasma stability through the introduction of a methyl group" did not find support in the subsequent experimental findings. The compound AVX420, used for further studies, exhibits a mere addition of one oxygen atom compared to AVX235. Thus, additional exploration is required to obtain further insights into the optimization of the compound structure.
4. The authors explain the non-selectivity of compounds with or without α -methyl for GVIA iPLA2 and GV sPLA2 with the same description, "MINIMAL inhibition" (on page 8 line 8 and line 16), but note that there is a disparity in inhibition rates between these two classes of compounds (Table 1).
5. Further pathway experiments and in vivo and PK studies are required to validate the mechanism and ascertain the effects.

Reviewer #2

(Remarks to the Author)

Johansen modified the structure of GK470, resulting in the improved activity of compound GK420. The related studies demonstrated that AVX420 induced the apoptosis in tumor cells through upregulation of intracellular oxidative stress response and transcriptional signaling pathways. However, this manuscript was deficient in the crucial animal experiments and pertinent preclinical study, such as pharmacokinetics and bioavailability. Hence, in my opinion, the manuscript did not meet the criteria for publication in Nature Communications.

Reviewer #3

(Remarks to the Author)

In this elegant work, Ashcroft et al describe the design, synthesis and characterization of a novel potent and selective cPLA2 inhibitor with anticancer properties. This paper is meticulously written and was a joy to read despite its complexity and vast amount of data. The novel inhibitor was designed and synthesized on the basis of an existing thiazolyl ketone inhibitor with proven efficacy and preclinical safety. The authors synthesized a series of related compounds and tested their selectivity for cPLA2 α (PLA2G4A) against two structurally different PLA2s using an established in vitro mixed micelle assay. The new compounds were found to be selective for the cPLA2 enzyme, presenting minimal inhibition for recombinant GV sPLA2 (PLA2G5) and GVIA PLA2 (PLA2G6). GK420 was selected as a new lead based on its human plasma stability, inhibitory potency in vitro as well as its potency to reduce IL1 β elicited AA-release from cultured cells. GK420 also proved effective in ex vivo assays, reducing LPS- and calcium ionophore-induced lipid mediator generation in human plasma and isolated

human PBMCs, respectively. The authors then screened 66 cancer cell lines for growth inhibitory effects of the inhibitor and discovered a specific vulnerability in acute leukemias. IN parallel they performed screens with the starting compound and with a structurally distinct cPLA2 inhibitor. They went on to compare the measured potencies of the cPLA2 inhibitors in suppressing cancer cell line growth against a reference library of known chemotherapeutics to predict their mechanism of action. This revealed a similarity between GK420 and some drugs targeting PRC2, which is involved in epigenetic regulation. The authors also found associations between cPLA2 inhibition and the MLL and ASXL2 cancer driver genes, further indicating links between cPLA2 inhibition and the control of histone methylation. Genome-wide CLE expression data was then used to associate drug sensitivity with basal transcript levels, which revealed additional links with PRC2 and other transcriptional repressors. The authors also found significant correlations between the expression of several genes important for oxidative defense mechanisms, such as the Nrf2 pathway, and the resistance to cPLA2 inhibition. After performing RNAseq and ribo-Seq analyses to analyse global transcriptional responses to cPLA2 inhibition in a selected cell line the authors found links with ATF4-dependent signaling, ER stress and mTOR signaling. In summary, based on their cancer cell line growth inhibitory screen, transcriptomics, and valuable in-depth in silico analyses of existing chemotherapeutic screening datasets, cancer mutational data, as well as transcriptomic and genomic databases, the authors have provided a wealth of data that provide an insight into the anticancer activities of GK420. Their tremendous work will undoubtedly be valuable to the research community. This work could be further strengthened if the suggested mechanistic links between cPLA2, cellular oxidative and ER stress defenses, and epigenetic transcriptional control are corroborated with mechanistic insights at the molecular level.

1. The new compounds were tested against GV and GVIA recombinant PLA2s using a valid vitro assay, showing a relatively low/minimal level of inhibition for these structurally distinct PLA2s. Nevertheless, given the number of existing PLA2s enzymes in the superfamily it is still possible that the compound inhibits other PLA2 isoforms and even that the inhibition profile for the tested ones is different when used in cells and tissues. Moreover, the compound could inhibit other unrelated enzymes when used in vivo. I wonder if the authors have compared the effects of GK420 (on cellular eicosanoid release and on cancer cell viability/proliferation) with the effects of gene manipulation of cPLA2? Does depletion of cPLA2 result in similar alterations in cell viability, ROS production, eicosanoid generation? This would help strengthen the idea that the identified mechanism of action of the compound in cancer cells is dependent on its selective inhibition of cPLA2.

2. The authors have convincingly demonstrated that GK420 induces ROS production and cell death in a number of hematological cancer cell lines. They describe the phenotypic and genetic changes leading to cancer cell death. In principle, as the authors state, cPLA2 could promote cancer cell proliferation through the mitogenic action of eicosanoids, such as PGE2. But cPLA2 also has some other functions, not related to lipid mediator production. Does the mechanism of action of GK420 on cancer cells depend on its capacity to inhibit cPLA2-induced AA release coupled with lipid mediator generation and signalling?

3. Combining their own screening data with available datasets the authors found links between the action of cPLA2 and epigenetic regulation. I wonder if genetic manipulation of cPLA2 in some of the identified sensitive cell lines would lead to alterations of the epigenetic mechanisms as predicted by these in silico analyses? It would also be very interesting to see if this potential epigenetic control is linked with the cPLA2-AA signaling axis or perhaps with other cPLA2 functions not related to lipid mediator signalling.

4. Correlations between cPLA2 inhibitor activity and gene expression data suggested that cPLA2-induced cancer cell death is linked with the induction of oxidative stress and that the resistance to the inhibitor could be mediated by Nrf2 signaling. Have the authors tried to target Nrf2 in combination with cPLA2 genetic manipulation in representative resistant vs sensitive cell lines to confirm the suggested mechanism? Does cPLA2 inhibition induce ER stress and alter Nrf2 signaling in relevant cell models? Do these events precede the measured rise in oxidative stress? Similarly, an important added value to this work would be if the authors could connect the dots between cPLA2 and ATF4, PRC2/mTOR signaling by cellular mechanistic studies using genetic manipulation of cPLA2 and analyses of ATF4 and mTOR/PRC2 signaling at the protein level.

Minor:

5. Figure S3 resolution should be improved.

Reviewer #4

(Remarks to the Author)

Review of NCOMMS-23-51095

"Next Generation Thiazolyl Ketone Inhibitors of Cytosolic Phospholipase A2 α : from Synthesis to Chemotherapeutic Mechanism of Action"

In this study, the teams of Professors Johansen and Kokotos performed the synthesis of AVX420, a potent and highly selective new generation thiazolyl ketone inhibitor of cPLA2 α using as a starting point the previously reported GK470 (AVX235) inhibitor. Using a number of complementary approaches, such as multiple cancer cell lines screening in combination with RNA-seq and Ribo-seq approaches the authors present data that suggests that AVX420 promotes selective growth inhibitory effects in cells of acute leukemias and other cancers both by increasing intracellular reactive oxygen species (ROS) and through a transcriptional reactivation of suppressed stress response genes resulting in increased cancer cell death. In addition, their studies point to a novel function of cPLA2 α in enabling cancer cell survival during oxidative stress by epigenetically controlling the expression of tumour suppressor genes.

This is an original and interesting paper. It explores the mechanism of action of the newly synthesized and previously reported inhibitors of cPLA2 α as targeted anti-cancer drugs and provides insights into the role of cPLA2 α in supporting cancer resistance, which will be of interest to the fields of cPLA2 α biology and cancer.

The article is characterized by clear and concise writing. It is methodologically sound and the experiments are carefully conducted and complementary and fully support the conclusions.

Data analysis is accurately performed and adequately described and data is clearly presented.

In general, I find this article as appropriate for publication in Nature Communication.

Major point

In two cases, the authors use correlations with an FDR of 5%, which is considered significant, and end up with a small number of cancer-related genes to be associated with sensitivity to AVX420 (Table S4). To increase the common associations between the cPLA2a inhibitors, they also use an FDR of 20%, which results in an increased number of associations, although these are clearly not significant. The authors are honest enough to clearly state it in the text, however they do provide a table (Table S5) with the additional identified associations and genes. Since the authors do not based their conclusions or their downstream analysis on these additional identified genes, I suggest to completely remove the analysis performed and the obtained data using an FDR of 20% as it does not add anything to their study.

Version 1:

Reviewer comments:

Reviewer #1

(Remarks to the Author)

Editorial note: Reviewer 3 has assessed authors responses to concerns of Reviewer 1, and considers them addressed

Reviewer #3

(Remarks to the Author)

In this revised version of the manuscript, the authors have conducted a number of pertinent additional experiments and have answered the key concerns raised in the original review. I recommend publication of this paper.

Reviewer #1

Comments:

Ashcroft et al. reported a new class of thiazolyl ketone-based inhibitors targeting cPLA2 α and explored their potential mechanism for chemotherapy. The following issues might be helpful to the authors:

1. The acquisition of crystal structures of protein-ligand complexes is a crucial aspect in the process of structure modification, as it provides essential binding information. Unfortunately, this study is deficient in the availability of pertinent data pertaining to crystal structures.

Response

The Editor stated that x-ray crystal structural work is not necessary for the revision. We concur because there is only one published crystal structure of cPLA2 from 25 years ago (Dessen et al, (1999) Ref 80 in the manuscript), and it has proved to be a very difficult enzyme to crystallize, let alone in the presence of an inhibitor.

2. The directing binding of AVX420 with cPLA2 α was not characterized.

Response

Molecular dynamics (MD) simulations followed by docking simulations have been carried out to characterize the interactions between GK420 and the active site of cPLA2 α . In addition, further plasma stability studies have been conducted. Utilizing LC-HRMS and an analytical approach to identify the suspected inhibitor and taking advantage of the high mass accuracy of HRMS, we now provide experimental evidence that GK420 acts as a prodrug, generating the corresponding acid, which binds to the active site of cPLA2 α via multiple interactions. Thus, a paragraph and a figure (Fig 3 added in the revised version) have been added to the main text and additional figures and movies have been added in the supplementary material.

“GK420 may act as a prodrug, generating the corresponding acid, which binds to the active site of cPLA2 α via hydrogen bonds and pi-pi interactions

To investigate the interactions between GK420 and the active site of cPLA2 α , we performed molecular dynamics (MD) simulations followed by docking simulations. However, the significant interactions between GK420 and amino acid residues were not observed in the docking simulations. Exploring the human plasma stability of GK420 by a LC-HRMS method, we envisioned that the ester group of GK420 might be hydrolyzed by cellular esterase's, generating the corresponding acid. Taking advantage of the high mass accuracy of HRMS and following our analytical approach to identify the suspected active inhibitor, we screened for an exact mass corresponding to the acid (M-H⁻ m/z 376.1224). Gratifyingly, we identified a peak which appears after 1 h of incubation and gradually increases. (Fig S2). Therefore, we performed docking simulations and MD simulations using the acid form of GK420 (GK420 acid). The interactions between Arg200 and the carboxyl group of GK420 acid were consistent in all docking poses showing lower docking scores (Fig. 3a). The MD simulation further

supported strong interaction between Arg200 and the carboxyl group showing very high occupancy of hydrogen bonds for the entire 200 ns MD simulation (128% in total of side chain NH) (Movie SX and Fig. 3b). In addition, the hydrogen bonds with Ala578 and Phe199 and the pi-pi interactions with Phe199 and Phe397 were also observed and further strengthened the interaction of the compound (Fig. 3c). Of special note, we performed three independent docking and MD simulations using different initial structures, and the interactions with the above residues were reproducible. Based on the above experimental and computational data, we propose that GK420 may act as a prodrug, generating the corresponding acid, which strongly interacts with the active site of cPLA2 α via multiple critical interactions.”

3. In Scheme 1A, the mention of a "potential increase in plasma stability through the introduction of a methyl group" did not find support in the subsequent experimental findings. The compound AVX420, used for further studies, exhibits a mere addition of one oxygen atom compared to AVX235. Thus, additional exploration is required to obtain further insights into the optimization of the compound structure.

Response

Unfortunately, the introduction of a methyl group did not lead to an increase of the plasma stability in this series of inhibitors. We have added a sentence commenting that. “in contrast to our expectations, in this series of thiazolyl ketones, the introduction of a methyl group did not offer greater stability.”

Although AVX420 differs by one oxygen atom compared to AVX235, AVX420 presented interesting properties, opening a new structural modification for application to PLA2 inhibitors.

4. The authors explain the non-selectivity of compounds with or without α -methyl for GVIA iPLA2 and GV sPLA2 with the same description, "MINIMAL inhibition" (on page 8 line 8 and line 16), but note that there is a disparity in inhibition rates between these two classes of compounds (Table 1).

Response

We have clarified this in the revised manuscript.

For compounds without a methyl group: “All the new analogs mentioned above can be considered selective inhibitors of cPLA2 α , because all present minimal inhibition (0-46%) of GVIA iPLA2 and GV sPLA2 at high concentrations (0.091 mole fraction), as shown in Table 1.”

For compounds with a methyl group: “Again, α -methyl substituted analogs can be considered selective inhibitors of cPLA2 α , because all present minimal inhibition (36-67%) of GVIA iPLA2 and GV sPLA2 at high concentrations (0.091 mole fraction) (Table 1).”

5. Further pathway experiments and in vivo and PK studies are required to validate the mechanism and ascertain the effects.

Response

Further pathway experiments have been conducted (see the response to reviewer 3).

PK study: As suggested by the Editor, a PK study is not a prerequisite for sending the manuscript back for re-review, and we feel that this would be beyond the scope of the current study, but would constitute a worthwhile further study for a future manuscript.

Reviewer #2

Comments

Johansen modified the structure of GK470, resulting in the improved activity of compound GK420. The related studies demonstrated that AVX420 induced the apoptosis in tumor cells through upregulation of intracellular oxidative stress response and transcriptional signaling pathways. However, this manuscript was deficient in the crucial animal experiments and pertinent preclinical study, such as pharmacokinetics and bioavailability. Hence, in my opinion, the manuscript did not meet the criteria for publication in Nature Communications.

Response

As suggested by the Editor, a PK study is not a prerequisite for sending the manuscript back for re-review, and we feel that this would be beyond the scope of the current study, but would constitute a worthwhile further study for a future manuscript.

A comparison of the stability of the different compounds in *in vitro* human plasma stability assays is shown in table 1. Although not indicative of *in vivo* stability and pharmacokinetics, these results, when taken together with the i) efficacy and specificity data shown in the same table and ii) the other results throughout the manuscript, show that AVX420 is a superior compound to GK470. The reviewer is correct in identifying a need to perform preclinical animal experiments to further characterize and compare the compounds. However, due to the lipid/hydrophobic like structure and inherent properties of the compounds that make them difficult to formulate, administer to animals, extract, and consistently analyze their pK profile in biological samples (e.g. solubility properties, sensitivity to oxidation/reduction, influence of different coagulants and extraction methods to compound isolation and stability, significant differences in levels achieved in animals after administration utilizing different routes that impact on bioavailability and others parameters), such experiments could not be concluded at the time of preparation of this manuscript (or even at this point, several months after the original submission). Although we do recognize that such experiments would be helpful to include in the manuscript, we believe that the lack of this data does not alter the key message of the manuscript revealed by our studies i.e. the identification of a novel mechanism of action and downstream targets of cPLA2a. Following the reviewer`s comment, a sentence has been added to **Discussion (page 26 line 559-560)** to indicate the importance of further preclinical development. **“Further preclinical experimentation, including *in vivo* stability and pharmacokinetics, will be critical next steps in the evaluation of the compound for clinical use.”**

Reviewer #3

Comments

In this elegant work, Ashcroft et al describe the design, synthesis and characterization of a novel potent and selective cPLA2 inhibitor with anticancer properties. This paper is meticulously written and was a joy to read despite its complexity and vast amount of data. The novel inhibitor was designed and synthesized on the basis of an existing thiazolyl ketone inhibitor with proven efficacy and preclinical safety. The authors synthesized a series of related compounds and tested their selectivity for cPLA2 α (PLA2G4A) against two structurally different PLA2s using an established in vitro mixed micelle assay. The new compounds were found to be selective for the cPLA2 enzyme, presenting minimal inhibition for recombinant GV sPLA2 (PLA2G5) and GVIA PLA2 (PLA2G6). GK420 was selected as a new lead based on its human plasma stability, inhibitory potency in vitro as well as its potency to reduce IL1 β elicited AA-release from cultured cells. GK420 also proved effective in ex vivo assays, reducing LPS- and calcium ionophore-induced lipid mediator generation in human plasma and isolated human PBMCs, respectively. The authors then screened 66 cancer cell lines for growth inhibitory effects of the inhibitor and discovered a specific vulnerability in acute leukemias. IN parallel they performed screens with the starting compound and with a structurally distinct cPLA2 inhibitor. They went on to compare the measured potencies of the cPLA2 inhibitors in suppressing cancer cell line growth against a reference library of known chemotherapeutics to predict their mechanism of action. This revealed a similarity between GK420 and some drugs targeting PRC2, which is involved in epigenetic regulation. The authors also found associations between cPLA2 inhibition and the MLL and ASXL2 cancer driver genes, further indicating links between cPLA2 inhibition and the control of histone methylation. Genome-wide CCLE expression data was then used to associate drug sensitivity with basal transcript levels, which revealed additional links with PRC2 and other transcriptional repressors. The authors also found significant correlations between the expression of several genes important for oxidative defense mechanisms, such as the Nrf2 pathway, and the resistance to cPLA2 inhibition. After performing RNAseq and ribo-Seq analyses to analyse global transcriptional responses to cPLA2 inhibition in a selected cell line the authors found links with ATF4-dependent signaling, ER stress and mTOR signaling. In summary, based on their cancer cell line growth inhibitory screen, transcriptomics, and valuable in-depth in silico analyses of existing chemotherapeutic screening datasets, cancer mutational data, as well as transcriptomic and genomic databases, the authors have provided a wealth of data that provide an insight into the anticancer activities of GK420. Their tremendous work will undoubtedly be valuable to the research community. This work could be further strengthened if the suggested mechanistic links between cPLA2, cellular oxidative and ER stress defenses, and epigenetic transcriptional control are corroborated with mechanistic insights at the molecular level.

Response

We very much appreciate this reviewer's understanding and very positive evaluation of the manuscript's importance as well as the suggestions for further improvement as addressed in the specific points below.

1. The new compounds were tested against GV and GVIA recombinant PLA2s using a valid *vitro* assay, showing a relatively low/minimal level of inhibition for these structurally distinct PLA2s. Nevertheless, given the number of existing PLA2s enzymes in the superfamily it is still possible that the compound inhibits other PLA2 isoforms and even that the inhibition profile for the tested ones is different when used in cells and tissues. Moreover, the compound could inhibit other unrelated enzymes when used *in vivo*. I wonder if the authors have compared the effects of GK420 (on cellular eicosanoid release and on cancer cell viability/proliferation) with the effects of gene manipulation of cPLA2? Does depletion of cPLA2 result in similar alterations in cell viability, ROS production, eicosanoid generation? This would help strengthen the idea that the identified mechanism of action of the compound in cancer cells is dependent on its selective inhibition of cPLA2.

Response

We address the question of whether genetic depletion of cPLA2 results in similar alterations in cell viability and ROS production with additional experiments presented in **Fig. 5 i-k**. Briefly, transient overexpression of a short hairpin (sh) RNA targeting PLA2G4A was used to achieve partial genetic knockdown compared to expression of a non-targeting control (NTC) sh-RNA. We saw higher levels of iROS and a trend towards lower cell viability that supported our findings regarding the role of cPLA₂α in preventing iROS accumulation and cell death in T-ALL.

We further support our proposal that the increase in iROS is specific to cPLA₂α inhibition using additional cPLA₂α inhibitors that are structurally unrelated to AVX420. These results are presented in **Fig. 5 h**.

Unfortunately, we experienced several **limitations to the use of genetic manipulation of cPLA₂α in T-ALL cell lines**, listed below and supported by data presented to the reviewers in Figure 1 below.

1. Consistent with other published studies, we experienced poor lentiviral transduction efficiencies in the CCRF-CEM cell line (average of **3% for cPLA2-sh** and 8% for NTC-sh). This was in comparison to >90% efficiency in the HEK-293T cells.
2. Cells transduced with the NTC-sh had significantly higher expression of PLA2G4A versus mock-transduced controls. cPLA2-sh expression reduced PLA2G4A expression versus the NTC-sh but remained **7-fold higher** than the mock-transduced cells.
3. Cells transduced with shRNA had **higher basal levels of iROS and cell death** versus the mock-transduced cells indicating the presence of basal state of oxidative stress.
4. We were unable to generate CCRF-CEM cells with stable knockdown of cPLA₂α. After an extended period of low cell viability, the antibiotic resistant populations from NTC-sh and cPLA2-sh transduction had equivalent expression of PLA2G4A.

Figure 2. Lentiviral expression system leads to induction of PLA2G4A expression and increased iROS. CCRF-CEM cells were transduced to express shRNA targeting PLA2G4A (cPLA2-sh) or a non-targeting controls shRNA (NTC-sh), mock transduced using polybrene alone (control). Transduction efficiencies measured by flow cytometry are shown in panel (i). Flow assisted cell sorting or flow cytometry based on GFP expression was used to isolate the transduced populations for analysis of (ii) expression of PLA2G4A, (iii) intracellular ROS, and (iv) cell death. Data are the mean of 2-4 replicates. One way ANOVA * $p < 0.05$, ** $p < 0.1$

We considered these findings major limitations to performing further cellular and mechanistic studies following genetic inhibition of cPLA2 in the T-ALL cell lines.

2. The authors have convincingly demonstrated that GK420 induces ROS production and cell death in a number of hematological cancer cell lines. They describe the phenotypic and genetic changes leading to cancer cell death. In principle, as the authors state, cPLA2 could promote cancer cell proliferation through the mitogenic action of eicosanoids, such as PGE2. But cPLA2 also has some other functions, not related to lipid mediator production. Does the mechanism of action of GK420 on cancer cells depend on its capacity to inhibit cPLA2-induced AA release coupled with lipid mediator generation and signaling?

Response

To address this important point, we included an exogenous source of AA during treatment with AVX420 and tested whether this prevented cell death, or the increased iROS induced by AVX420. The data are presented in **Fig 5f** and **Fig. S8** and suggest that neither AA nor eicosanoid metabolites could prevent cell death or iROS induced by AVX420. Rather we found that exogenous AA augmented the loss of viability resulting from AVX420 treatment, indicating cPLA2 activity may protect cells from an abundance of free AA in this cancer type.

Additionally, we attempted to identify eicosanoids released by the CCRF-CEM cells using HPLC-MS. Unlike in whole blood and PBMCs however, we were not able to reliably identify any eicosanoids released from the T-ALL cell line. We looked for 6-keto prostaglandin F1 α , thromboxane B2, prostaglandin F2 α , prostaglandin E2, prostaglandin D2, 12(S)-HHTrE, 15(S)-HETE, 12(S)-HETE, 5(S)-HETE, and LTB4, in both control conditions, and in response to the calcium ionophore A23187.

While this is not an exhaustive study, our data suggests an alternative function of cPLA2 in this cancer, which as suggested by the reviewer, could be responsible for the effects of AVX420 in this case. We have revised **figure 6d** and altered the **text on page 23, line 467-476** to be in line with these findings and the reviewers' comments.

“Given we observed increased iROS 1 hour after treatment with AVX420 (Fig S8a), before both the transcriptional response and cell death (Fig S8b) and that exogenous AA was ineffective at rescuing these responses (Figs. 5e and S8c), we propose that metabolic alterations could be the cause of the oxidative stress. In addition to its central role in eicosanoid metabolism cPLA2 participates in phospholipid (PL) remodeling via the Lands Cycle, and is important for the formation of lipid droplets, which are a significant fuel source in certain cancers including acute leukemias. We thus hypothesize that cPLA2 may support FAO, which in addition to providing energy protects cells against oxidative stress via generation of NADPH. Inhibition of cPLA2 would then have both epigenetic and metabolic consequences for this cell type (**Fig. 6d**).”

3. Combining their own screening data with available datasets the authors found links between the action of cPLA2 and epigenetic regulation. I wonder if genetic manipulation of cPLA2 in some of the identified sensitive cell lines would lead to alterations of the epigenetic mechanisms as predicted by these *in silico* analyses? It would also be very interesting to see if this potential epigenetic control is linked with the cPLA2-AA signaling axis or perhaps with other cPLA2 functions not related to lipid mediator signaling.

Response

Both our *in silico* and transcriptomic analyses pointed to a role for cPLA2 signaling in the epigenetic suppression of ATF4-dependent gene transcription. Due to the limited cellular material available (as discussed under point#1), we attempted to investigate whether genetic manipulation of cPLA2 affected transcript levels of three ATF4 target genes (ATF3, SESN2 and CHAC1), and also 1 Nrf2 target gene (SLCA11). RNA was isolated from transduced cells collected by FACS. Interestingly, both NTC-sh and cPLA2-sh expressing cells had slightly lower expression of the ATF4 target genes versus mock-transduced control (Figure 2 below). This was surprising given the observed increase in iROS in these cells (see figure 1 (iii) above) and was in contrast to the Nrf2 target gene which was increased in both cell populations. In line with our proposed mechanism, this could be explained by epigenetic suppression of ATF4 target gene transcription resulting from the increased expression of cPLA2 observed in both transduced populations versus the mock transfected cells (Figure 1 (ii) above). Since this a 'side-effect' of the experimental approach, we have not included the data in the revised version of the manuscript.

Figure 2 Expression of ATF4 and Nrf2 target genes measured by q-RT-PCR. Data are expressed relative to the mock-transfected control (dotted line).

4. Correlations between cPLA2 inhibitor activity and gene expression data suggested that cPLA2-induced cancer cell death is linked with the induction of oxidative stress and that the resistance to the inhibitor could be mediated by Nrf2 signaling. Have the authors tried to target Nrf2 in combination with cPLA2 genetic manipulation in representative resistant vs sensitive cell lines to confirm the suggested mechanism? Does cPLA2 inhibition induce ER stress and alter Nrf2 signaling in relevant cell models? Do these events precede the measured rise in oxidative stress? Similarly, an important added value to this work would be if the authors could connect the dots between cPLA2 and ATF4, PRC2/mTOR signaling by cellular mechanistic studies using genetic manipulation of cPLA2 and analyses of ATF4 and mTOR/PRC2 signaling at the protein level.

Response

While we agree in principle that combining genetic manipulation of Nrf2 with cPLA2 and additional cellular mechanistic studies using genetic manipulation of cPLA2 and analysis of ATF4, mTOR/PRC2 at the protein level would provide additional valuable insights into the mechanism of action of AVX420, all of the practical limitations to genetic manipulation of cPLA2 in T-ALL cells listed under point #1 would need to be addressed before embarking on these experiments. This makes it unrealistic for us to achieve these goals within the scope of this study and as of now, remains a challenge for the future.

Regarding the question of whether cPLA2 inhibition induces ER stress in relevant cell models, we point to our analysis of the RNA-SEQ and Ribo-SEQ data in CCRF-CEM cells, where we concluded that ATF4 dependent transcription in response to AV420 did most likely not result from a classic ER stress response. **See page 22- lines 437-444.** “Close inspection of the genes upregulated after 6 hours treatment with AVX420 revealed that many are established targets of ATF4 (E.g., CHAC1, ADM2, SESN2, SLC7A11, DDIT3, ULBP1, ATF3) and GO term and KEGG pathway enrichment analysis performed on the 82 DEGs at 16 hours predicted a response to endoplasmic reticulum (ER) stress (**Table S10**). While these findings supported the hypothesis that AVX420 initiated cell death via the ATF4 dependent ISR pathway, the absence of translational repression, or selective upregulation of ATF4 translation in response to AVX420,

would strongly argue against this, and an alternative mechanism for activation of ATF4 target genes is more likely.”

Regarding the question of whether cPLA2 inhibition alters Nrf2 signaling in relevant cell models, we point again to our analysis of the RNA-SEQ data where we showed significant overlap between AVX420 regulated transcripts and Nrf2-target genes. This would be consistent with activation of Nrf-2 signaling in response to the increased intracellular ROS seen in response to AVX420. **See page 23 line 458-461** *“We also found significant overlaps with Nrf2-dependent genes (Fig. 6b), and predicted targets of the transcription factors MafG, Nrf1 and Hes2 (Fig. 6c). Both MafG and Nrf1 are known to cooperate with Nrf2 to regulate the expression of antioxidant response genes...”*

Regarding whether transcriptional events precede the measured rise in oxidative stress, we showed a time course for the induction of ROS (now Fig. S8 in the revised version of the manuscript). This can be compared to the time course for transcriptional changes as measured by RNA-SEQ. Increased iROS was seen as early as 1 hour after treatment with AVX420, peaking 8-12 hours after treatment. In contrast, we found no differentially expressed transcripts by RNA-SEQ or Ribo-SEQ at 1 hour after treatment with AVX420. Differential gene expression was observed after 6 hours treatment while loss of cell viability only began 8 hours after treatment. The timing of these events implies that the increase in iROS initiated alteration to expression of target genes that were subsequently involved in activation of cell death.

These findings are highlighted and clarified in the altered text on page 23 line 467-468 of the revised manuscript.

“Given we observed increased iROS 1 hour after treatment with AVX420 (Fig S8a), before both the transcriptional response and cell death (Fig S8b)...”

5. Figure S3 resolution should be improved.

Response

Figure S3 (figure S4 in the revised version) will be provided in a display format for viewers to be able to read the text clearly.

Reviewer # 4

Comments

In this study, the teams of Professors Johansen and Kokotos performed the synthesis of AVX420, a potent and highly selective new generation thiazolyl ketone inhibitor of cPLA2 α using as a starting point the previously reported GK470 (AVX235) inhibitor. Using a number of complementary approaches, such as multiple cancer cell lines screening in combination with RNA-seq and Ribo-seq approaches the authors present data that suggests that AVX420 promotes selective growth inhibitory effects in cells of acute leukemias and other cancers both by increasing intracellular reactive oxygen species (ROS) and through a transcriptional reactivation of suppressed stress response genes resulting in increased cancer cell death. In

addition, their studies point to a novel function of cPLA2 α in enabling cancer cell survival during oxidative stress by epigenetically controlling the expression of tumour suppressor genes.

This is an original and interesting paper. It explores the mechanism of action of the newly synthesized and previously reported inhibitors of cPLA2 α as targeted anti-cancer drugs and provides insights into the role of cPLA2 α in supporting cancer resistance, which will be of interest to the fields of cPLA2 α biology and cancer.

The article is characterized by clear and concise writing. It is methodologically sound and the experiments are carefully conducted and complementary and fully support the conclusions. Data analysis is accurately performed and adequately described and data is clearly presented. In general, I find this article as appropriate for publication in Nature Communication.

We very much appreciate this reviewer's very positive evaluation of the manuscript's importance and appropriateness for publication in Nature Communications.

1. In two cases, the authors use correlations with an FDR of 5%, which is considered significant, and end up with a small number of cancer-related genes to be associated with sensitivity to AVX420 (Table S4). To increase the common associations between the cPLA2 α inhibitors, they also use an FDR of 20%, which results in an increased number of associations, although these are clearly not significant. The authors are honest enough to clearly state it in the text, however they do provide a table (Table S5) with the additional identified associations and genes. Since the authors do not based their conclusions or their downstream analysis on these additional identified genes, I suggest to completely remove the analysis performed and the obtained data using an FDR of 20% as it does not add anything to their study.

Response

In line with the reviewer's request, we have removed the data obtained using an FDR of 20% (table S5 and page 17 line 344).

Additional alterations to the manuscript

Methods

1. Inclusion of methods for molecular docking and molecular dynamics (MD) simulations
2. Inclusion of methods for genetic knockdown of cPLA2 and validation
3. Removed part of RNA-SEQ and Ribo-SEQ methods and moved to Supplemental Methods

Supporting data

1. Added Fig S2. LC-HRMS analysis of AVX420 in human plasma.
2. Added Fig. S8c